# Integration of microbattery with thin-film electronics for constructing an integrated transparent microsystem based on InGaZnO

Bin Jia [1], Chao Zhang[1], Min Liu[1], Zhen Li[1], Jian Wang[1], Li Zhong [1], Chuanyu Han [2], Ming Qin[1] & Xiaodong Huang[1] ✉

A full integration of miniaturized transparent energy device (lithium-ion battery), electronic device (thin-film transistor) and sensing device (photodetector) to form a monolithic integrated microsystem greatly enhances the functions of transparent electronics. Here, InGaZnO is explored to prepare the above devices and microsystem due to its multifunctional properties. A transparent lithium-ion battery with InGaZnO as anode (capacity~9.8 µAh $cm^{-2}$) is proposed as the on-chip power source. Then, thin-film transistor with InGaZnO as channel (mobility~23.3 $cm^2\ V^{-1}\ s^{-1}$) and photodetector with InGaZnO as photosensitive layer (responsivity~0.35 A $W^{-1}$) are also prepared on the substrate for constructing an fully integrated transparent microsystem. Each device displays acceptable performance. Moreover, alternating-current signals can be successfully charged into the lithium-ion battery by using the thin-film transistor as the on-chip rectifier and also the photodetector works well by using the charged battery as the on-chip power, demonstrating collaborative capabilities of each device to achieve systematic functions.

Transparent electronics have promising applications in the next-generation consumer electronics (e.g., transparent displays, mobile phones, and laptops), smart home and transportation (e.g., transparent solar cells and smart windows), and securities (e.g., invisible cameras and detectors)[1–3]. Enhancing the functions is one of the main trends driving the development of transparent electronics. This can be achieved by integrating other kinds of components (e.g., sensors and lithium-ion batteries, LIBs) with electronic devices (e.g., thin-film transistors, TFTs) for constructing a microsystem, which is also known as the More than Moore strategy[4–6]. In this microsystem, LIB works as the power source for other components (e.g., sensors and TFTs). Particularly, for portable transparent electronics (e.g., transparent mobile phones and wireless detectors), it is difficult to use electrical wirings for power supply and thus LIB is essential in these applications. Integration of various components in one substrate (called as monolithic integration) can make the microsystem more compact and lightweight[7,8]. Moreover, monolithic integration is effective to improve

working speed and suppress power consumption by reducing the connection length and relevant parasitic effects between each component. However, an integrated transparent microsystem including the above-mentioned essential components has not been realized yet. In addition, constructing an integrated microsystem based on the same functional material is helpful and desirable to enhance the microsystem compactness and simplify its fabrication process. Due to good uniformity, high carrier mobility and high optically transparency, amorphous InGaZnO (IGZO) film has received much attention in recent years[9–11]. So far, IGZO has already been used in transparent electronic devices (e.g., TFT) and sensors (e.g., photodetector, PD), respectively[12–15]; on the other hand, there is still lack of IGZO-based transparent energy devices for constructing a fully integrated transparent microsystem.

In this work, a transparent thin-film lithium-ion battery (LIB) with IGZO as the anode is proposed as the on-chip power source. Then, TFT with IGZO as the channel layer and PD with IGZO as the photosensitive

[1]Key Laboratory of MEMS of the Ministry of Education, School of Integrated Circuits, Southeast University, Nanjing 210096, China. [2]School of Microelectronics, Faculty of Electronics and Information, Xi'an Jiaotong University, Xi'an 710049, China. ✉e-mail: xdhuang@seu.edu.cn

layer are also prepared. All the devices are fabricated on a single glass substrate for constructing an integrated transparent microsystem. An integrated transparent microsystem including all the above-mentioned essential components is realized in this work. It is demonstrated that each device displays an acceptable performance and collaboratively works well. Moreover, because all the essential devices use the same material IGZO as their functional layers, the fabrication and integration of this microsystem can be significantly simplified.

## Results

### Structure of the microsystem and its components

Figure 1a shows the schematic diagrams of this integrated transparent microsystem and each component. Figure 1b shows the corresponding equivalent circuit diagram of the integrated microsystem. Figure 1c displays the photograph of the complete integrated microsystem. The connections between each component in this integrated microsystem are realized by using ITO interconnects. LIB consists of ITO current collector|$V_2O_5$ cathode|LiPON electrolyte|IGZO anode|ITO current collector, where $V_2O_5$ and LiPON are chose as the cathode and electrolyte respectively mainly due to their high transparency, compatibility with the microelectronic process and good electrochemical properties in the amorphous state[16–19]. TFT consists of ITO gate/HfLaO dielectric/IGZO channel/ITO source and drain, and high-$k$ HfLaO is used as the dielectric mainly because of its high transparency, high crystalline temperature and effectiveness in reducing the operating voltages[20,21]. PD herein displays a photoresistive structure and consists of ITO electrode/IGZO photosensitive layer/ITO electrode. As demonstrated in Fig. 1d, each component as well as the integrated microsystem shows high transparency (>66% at 550-nm light) due to the rational material design. The detailed fabrication processes are shown in Supplementary Fig. 1 and Supplementary Table 1. The fabrication is performed at room temperature to ensure that the prepared films are amorphous, which is desirable to address the nonuniformity

issues of a microsystem caused by random grain boundaries in the films[22].

### Material and electrochemical properties of IGZO thin films

Although no work has been performed to explore the feasibility of IGZO used in LIB, its main compositions (ZnO, $In_2O_3$ and $Ga_2O_3$) have been demonstrated to be typical LIB anode materials with relatively high specific capacities, suggesting great potential of IGZO as the LIB anode[23–28]. The anode electrochemical performance is greatly affected by its electrical and ionic conductivities, which can be modulated by changing the oxygen content in IGZO. Therefore, IGZO films with different oxygen contents are prepared by changing the oxygen partial pressure ($P_{O2}$ = 0 Pa, 0.04 Pa and 0.15 Pa) during sputtering. Supplementary Fig. 2a shows the Raman spectra of the IGZO films deposited at different $P_{O2}$. All the films present typical characteristics of IGZO, demonstrating the IGZO formation[29–31]. The film compositions can be quantitatively characterized by using EDX (Supplementary Fig. 2b) and the relative elemental contents of the films are summarized in Supplementary Table 2. The oxygen content increases with increasing $P_{O2}$ and this suggests that the oxygen content in the IGZO film can be effectively modulated simply by changing $P_{O2}$. Moreover, it is found that each element is evenly distributed for the IGZO films (Supplementary Fig. 3). The Hall carrier concentration and mobility of the IGZO films prepared at different $P_{O2}$ have been measured by using a Hall-effect measurement system (Supplementary Fig. 2c). The Hall carrier concentration and mobility decrease with increasing $P_{O2}$. It is known that oxygen vacancy ($V_O$) is the main contributor to the electron carriers and electrical conductivity of IGZO[32]. Increasing $P_{O2}$ enhances the oxygen content (and thus suppresses the $V_O$ content) in IGZO, thus reducing the Hall carrier concentration and mobility. The crystallinity of the films is investigated by XRD (Supplementary Fig. 2d) and TEM (Supplementary Fig. 4), both of which suggest that all the films display an amorphous state. This is quite desirable for improving the uniformity of a microsystem.

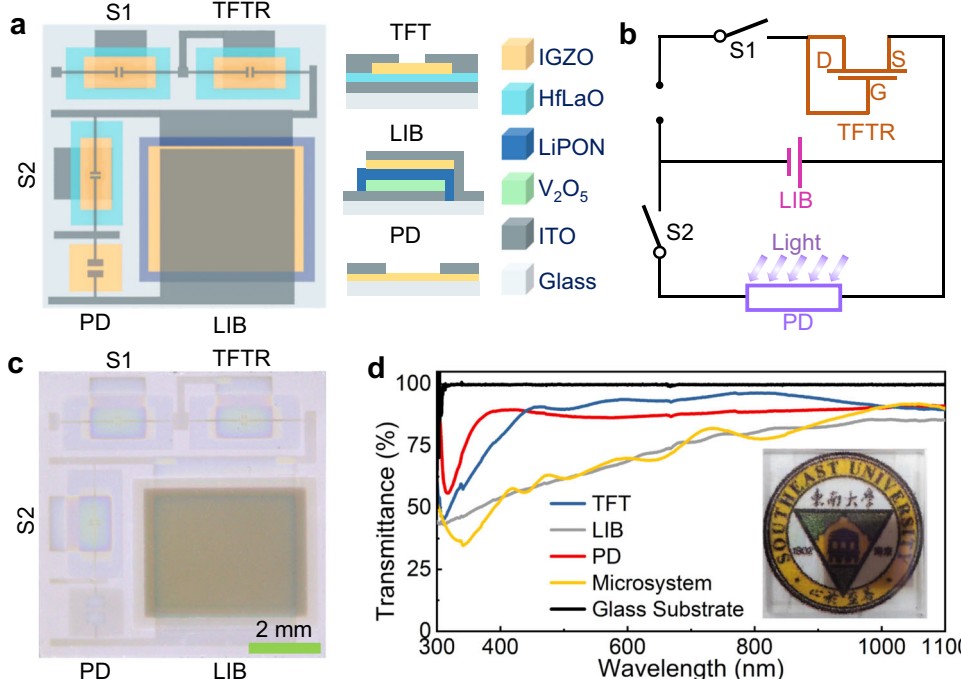

**Fig. 1 | Illustration of the integrated transparent microsystem as well as each component. a** Schematic diagrams of the integrated microsystem and its components. **b** Equivalent circuit diagram of the integrated microsystem. **c** Photograph of the complete integrated microsystem. **d** Transmittance of each component and the integrated microsystem. The inset is the optical image of the complete integrated microsystem placed onto a logo.

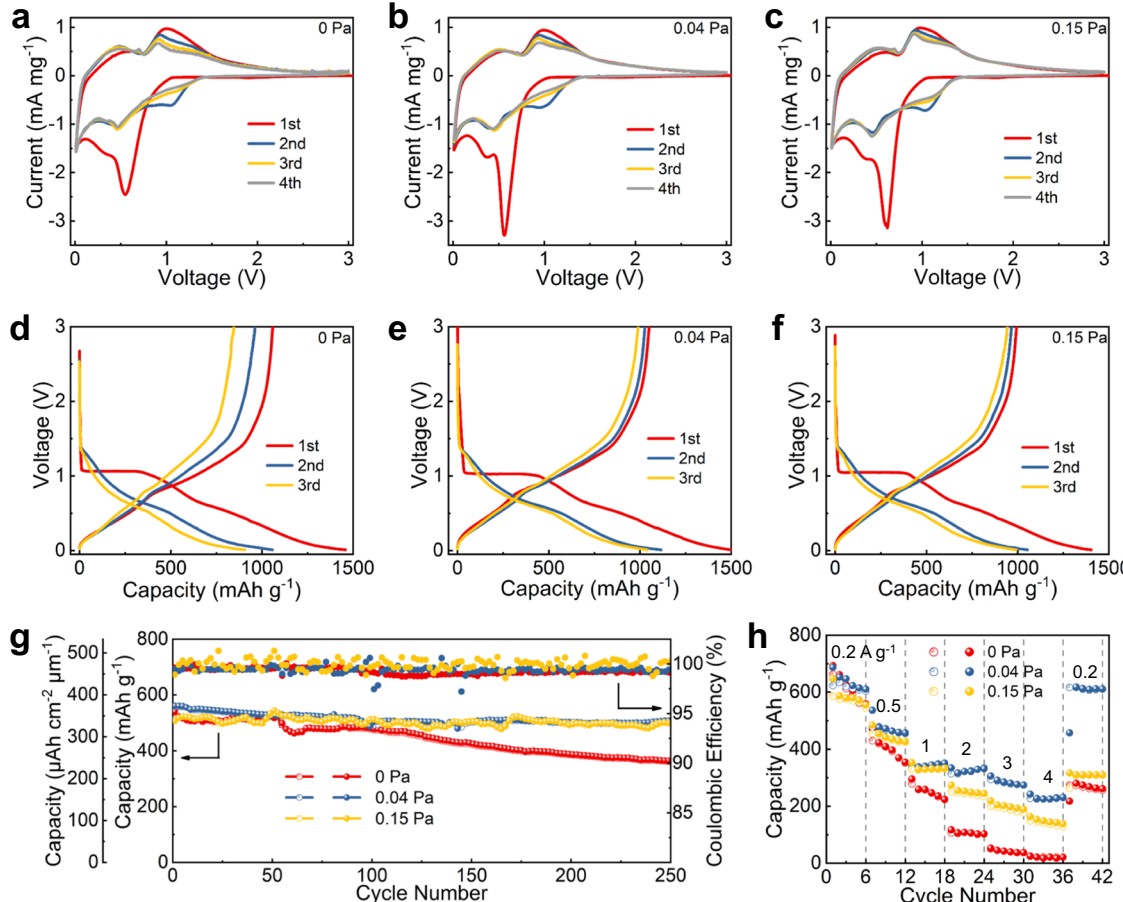

**Fig. 2 | Electrochemical performance of the 400-nm IGZO anode films prepared at different P$_{O2}$ measured based on the coin half cells. a–c** CV curves, **d–f** GCD curves, **g** cycling curves, and **h** rate performance. (Open symbol – charge; solid symbol – discharge.).

As a promising anode material, the electrochemical energy-storage characteristics of the IGZO films are firstly investigated based on the coin half cells. Figure 2a–c shows the cyclic voltammetry (CV) curves in a voltage range of 0.01–3.0 V at a scanning rate of 0.3 mV s$^{-1}$ for the samples prepared at various P$_{O2}$. All the samples display a cathodic peak at 0.6 V in the initial two cycles, which are mainly derived from the formation of a solid electrolyte interphase (SEI)[24,25,33]. In the subsequent cycles, the CV curves are reproducible, suggesting high reversibility of the following reactions. It is further found that the anodic (or cathodic) peaks in the subsequent CV curves are consistent well with the multistep alloying (or dealloying) processes of M (M = In, Ga and Zn metal) with Li$^{+}$. Because these alloying (or dealloying) processes occur in a similar potential range, thus resulting in relatively broad anodic (or cathodic) peaks in the CV curves. Based on the above results, the electrochemical processes of IGZO with Li$^{+}$ can be described by Supplementary Note 1, Supplementary Equations (1) and (2)[23–25,27,28]. The formation of Li$_2$O can effectively enhance the anode ionic conductivity[34,35]. In addition, Li$_2$O can also act as a buffer matrix to release the stress induced by the anode volume change during the lithiation/delithiation[34,36].

Figure 2d–f depicts the galvanostatic charge/discharge (GCD) voltage profiles of the samples between 0.01 and 3.0 V at a relatively low specific current of 0.05 A g$^{-1}$. All the samples display an obvious voltage plateau at around 1.0 V in the 1$^{st}$ discharging process. This is mainly ascribed to the SEI formation and thus results in a relatively low initial Coulombic efficiency (CE ~ 72.6%, 70.0%, 70.8% for the anodes prepared at P$_{O2}$ = 0 Pa, 0.04 Pa and 0.15 Pa, respectively). The specific capacity in the 1st process is about 1059, 1049, and 993 mAh g$^{-1}$ for the anodes prepared at P$_{O2}$ = 0, 0.04 and 0.15 Pa, respectively, which are a

little smaller than the IGZO theoretic value (-1311 mAh g$^{-1}$. The detailed calculation process is shown in Supplementary Note 1). The electrochemical reactions enhance with decreasing the charge/discharge current, thus helping make the experimental capacity closer to the theoretic value. In addition, the GCD curves of the anodes at P$_{O2}$ = 0.04 Pa and 0.15 Pa tend to coincide in the following cycles; for comparison, the GCD curves of the anode at P$_{O2}$ = 0 Pa degrades monotonically. Figure 2g shows the cycling characteristics of the samples at a specific current of 0.4 A g$^{-1}$. The retained reversible capacity after 250 cycles is 361 mAh g$^{-1}$ (or 241 µAh cm$^{-2}$ µm$^{-1}$), 505 mAh g$^{-1}$ (or 337 µAh cm$^{-2}$ µm$^{-1}$) and 497 mAh g$^{-1}$ (or 331 µAh cm$^{-2}$ µm$^{-1}$) for the IGZO anodes prepared at P$_{O2}$ = 0 Pa, 0.04 Pa and 0.15 Pa, corresponding to capacity retention of 67.1%, 91.5% and 94.8%, respectively. The IGZO anodes prepared at P$_{O2}$ = 0.04 Pa and 0.15 Pa present better cycling performance than the one at P$_{O2}$ = 0 Pa mainly due to their more Li$_2$O formation, which effectively acts as a buffer matrix for stress relaxation. Supplementary Fig. 5 presents the SEM images of the IGZO anodes before and after cycling. Different from the one at P$_{O2}$ = 0 Pa which cracks into pieces, the IGZO anode prepared at P$_{O2}$ = 0.04 Pa keeps intact after cycling, further demonstrating its stronger mechanical strength against the stress. Moreover, as seen in Supplementary Fig. 6, the anode at P$_{O2}$ = 0.04 Pa can work well even when the IGZO thickness increases from 400 to 800 nm, and the reversible capacity after 250 cycles retains 450 mAh g$^{-1}$ (or 300 µAh cm$^{-2}$ µm$^{-1}$), corresponding to capacity retention of 83.6%.

Figure 2h shows the rate capacities of the samples. For the IGZO anode prepared at P$_{O2}$ = 0.04 Pa, 36.4% (-227 mAh g$^{-1}$) of its capacity (relative to the initial capacity at 0.2 A g$^{-1}$) is obtained at a high specific current of 4 A g$^{-1}$. Also, as the specific current goes back to 0.2 A g$^{-1}$,

98.9% (-616 mAh g$^{-1}$) of its capacity can be well recovered. On the contrary, for the anode at P$_{O2}$ = 0 Pa (or the anode at P$_{O2}$ = 0.15 Pa), 3.7% (or 25.3%) of its capacity is obtained at 4 A g$^{-1}$ and only 41.3% (or 53.0%) of its capacity is left as the specific current returns to 0.2 A g$^{-1}$. It is obvious that the anode at P$_{O2}$ = 0.04 Pa displays superior rate performance than the other ones, suggesting its faster reaction kinetics. This can be further confirmed by the EIS spectra shown in Supplementary Fig. 7, which presents that the anode at P$_{O2}$ = 0.04 Pa shows the smallest charge-transfer resistance among the samples. The faster kinetics of the anode prepared at P$_{O2}$ = 0.04 Pa should be mainly because of its higher ionic conductivity caused by its sufficient Li$_2$O formation (vs. the anode at P$_{O2}$ = 0 Pa) and higher electrical conductivity (vs. the anode at P$_{O2}$ = 0.15 Pa). The critical electrochemical performance of the IGZO anodes at various P$_{O2}$ is summarized in Supplementary Table 4. Based on the above results, The IGZO film prepared at an oxygen partial pressure of 0.04 Pa shows better overall electrochemical performance than the other ones in terms of its higher reversible capacity, better cycling stability and better rate characteristics. As shown in Supplementary Table 5, the IGZO anode film in this work presents comparable performance to other typical metal-oxide anode films reported in the literature.

## All-solid-state thin-film transparent LIB with IGZO as the anode

After systematically investigating the IGZO electrochemical characteristics, an all-solid-state thin-film transparent LIB with IGZO anode is developed as the on-chip power source. As seen in Fig. 3a, this LIB consists of ITO electrode|IGZO anode |LiPON electrolyte|V$_2$O$_5$ cathode|ITO electrode and each layer can be clearly defined with a sharp and even interface. In addition to the IGZO film, both the electrolyte and cathode films also display an amorphous state (Supplementary Fig. 8), which is beneficial for improving the device and microsystem uniformities. It is known that mechanical stress would be induced by the volume change of the LIB anode during the lithiation/delithiation, and the induced stress increases with increasing the anode thickness,

thus a relatively thick IGZO is used in the half cells, which is more effective and suitable to examine its energy-storage characteristics. For comparison, a relatively thin IGZO is used in the thin-film LIB, TFT and PD mainly for improving the fabrication efficiency. Figure 3b shows the charge/discharge voltage profiles of the thin-film LIB between 0.5 and 3.0 V at a low current density of 1 μA cm$^{-2}$. The initial CE is about 44.3% and increases obviously in the following cycles. Figure 3c shows the LIB voltage profiles under various current densities. A relatively high reversible capacity of 9.8 μAh cm$^{-2}$ is achieved at 1 μA cm$^{-2}$. Moreover, a specific capacity of 4.8 μAh cm$^{-2}$ with capacity degradation of 51.0% can be still obtained even at a high current density of 14 μA cm$^{-2}$, suggesting its good rate characteristics. The V$_2$O$_5$-based thin-film LIB in this work has comparable performance to those in the literature in terms of its relatively high capacity and good cycling stability, as seen in Supplementary Table 6.

In order to investigate an optimum match between the cathode and anode, thin-film LIBs with different IGZO thicknesses while all the other layers maintaining constant are also prepared. As seen in Supplementary Fig. 9, the discharge capacity of the LIB with 40-nm IGZO anode is 5.7 μAh cm$^{-2}$ at 1 μA cm$^{-2}$ and degrades to 2.1 μAh cm$^{-2}$ at 14 μA cm$^{-2}$, corresponding to 63.2% degradation. The capacity of the LIB with 120-nm IGZO anode is 9.7 μAh cm$^{-2}$ at 1 μA cm$^{-2}$ and reduces to 4.1 μAh cm$^{-2}$ at 14 μA cm$^{-2}$, corresponding to 57.7% degradation. It is obvious that the LIB with 80-nm IGZO presents higher reversible capacity and better rate performance than the other ones, suggesting that the anode is well matched with the cathode in this case.

In practical use, devices are usually required to operate at high temperatures. According to the conventions, the devices need to enable work at 85 °C in the industrial electronics and at 125 °C in the military electronics. Figure 3d shows the LIB voltage profiles at different temperatures. The LIB displays a reliable performance even at 125 °C. In addition, it can be continually charged and discharged for more than 60 h at 85 °C with only a slight capacity loss of 11.3%, as seen in Supplementary Fig. 10. It is noted that metallic Li is commonly

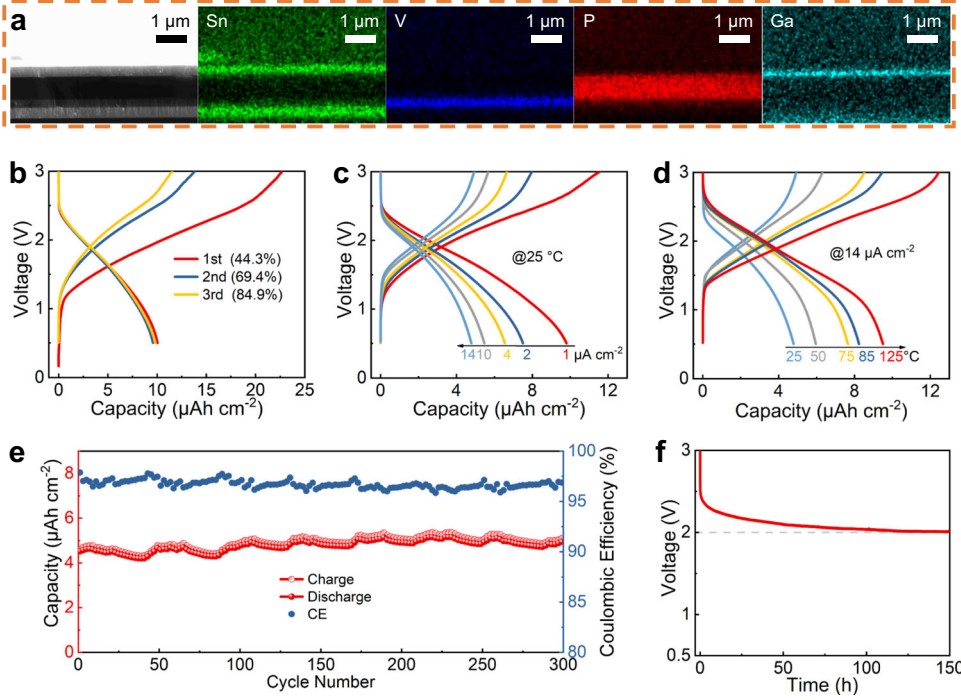

**Fig. 3 | Characterization of the all-solid-state thin-film LIB with 80-nm IGZO as the anode. a** Cross-sectional SEM image as well as the EDX elemental mappings. GCD curves of the LIB (**b**) for the initial three cycles at 1 μA cm$^{-2}$ and 25 °C, **c** under different current densities at 25 °C, and **d** under different temperatures at 14 μA cm$^{-2}$. **e** Cycling performance of the LIB at 14 μA cm$^{-2}$ and 25 °C. **f** Self-discharge curve of the LIB at 25 °C. The LIBs are activated at a low current density of 1 μA cm$^{-2}$ for three cycles firstly before testing in Fig. 3c–f.

adopted as the anode in the thin-film LIBs. The low melting temperature (-181 °C) of Li causes that the corresponding LIB is difficult to operate reliably in high temperatures. For comparison, IGZO as the anode not only contributes to the LIB transparency but also improves the LIB operating temperature in this work. Figure 3e exhibits the cycling characteristics of this thin-film LIB. Its capacity has no degradation even over 300 cycles and also a relatively high CE (≥ 96.5%) is retained during cycling, both of which demonstrate good cycling performance of this LIB. Figure 3f shows the self-discharge curve of the LIB. Once the LIB is charged to 3 V, the self-discharge process versus time is recorded and a quite low decay rate of 6.7 mV h$^{-1}$ is achieved for this LIB, demonstrating its good energy-storage capability.

**TFT with IGZO as the channel layer**

TFT is one main kind of electronic devices and its main function in a microsystem includes switching and driving. In this work, TFT with a bottom-gate and top-contact structure is prepared (Fig. 1a) and it consists of ITO gate/HfLaO dielectric/IGZO channel/ITO source and drain. The HfLaO dielectric displays an amorphous state (Supplementary Fig. 11), and thus is beneficial for improving the device and microsystem uniformities. The inset of Fig. 4a shows the areal capacitance and leakage as a function of applied voltage for the

HfLaO-based capacitor. The $k$ value of the dielectric is calculated to be 14.5. Also, the dielectric shows a low leakage (~7.6 × 10$^{-9}$ A or 10$^{-5}$ A cm$^{-2}$ at −1 V). The effects of IGZO prepared at various $P_{O2}$ on the TFT performance are also investigated. Figure 4a and Supplementary Fig. 12 show the transfer curves of the TFTs. The key electrical parameters, including the saturated carrier mobility ($\mu_{sat}$), subthreshold swing ($SS$), threshold voltage ($V_{th}$), hysteresis ($\Delta V_{th}$) and on-off current ratio ($I_{on}/I_{off}$), can be extracted from the transfer curves and summarized in Supplementary Table 4. The TFT at $P_{O2} = 0.04$ Pa displays higher $\mu_{sat}$ and larger $I_{on}/I_{off}$ than that at $P_{O2} = 0.15$ Pa mainly because of its more $V_O$ content and thus higher carrier density as demonstrated in Supplementary Fig. 2c. However, excessive $V_O$ content results in excessive traps in the IGZO film, which causes that the TFT at $P_{O2} = 0$ Pa presents much larger $\Delta V_{th}$ and larger $SS$ than the one at $P_{O2} = 0.04$ Pa. The TFT at $P_{O2} = 0.15$ Pa shows smaller $I_{on}/I_{off}$ than the TFTs at $P_{O2} = 0$ Pa and $P_{O2} = 0.04$ Pa mainly because of its much lower $\mu_{sat}$ and thus smaller on-current. Figure 4b presents the output characteristics of the TFT at $P_{O2} = 0.04$ Pa and a driving output current $I_D$ of 10.5 μA is obtained at $V_{DS} = 5$ V and $V_{GS} = 5$ V. This TFT exhibits an acceptable performance even prepared at room temperature and without any thermal annealing treatment.

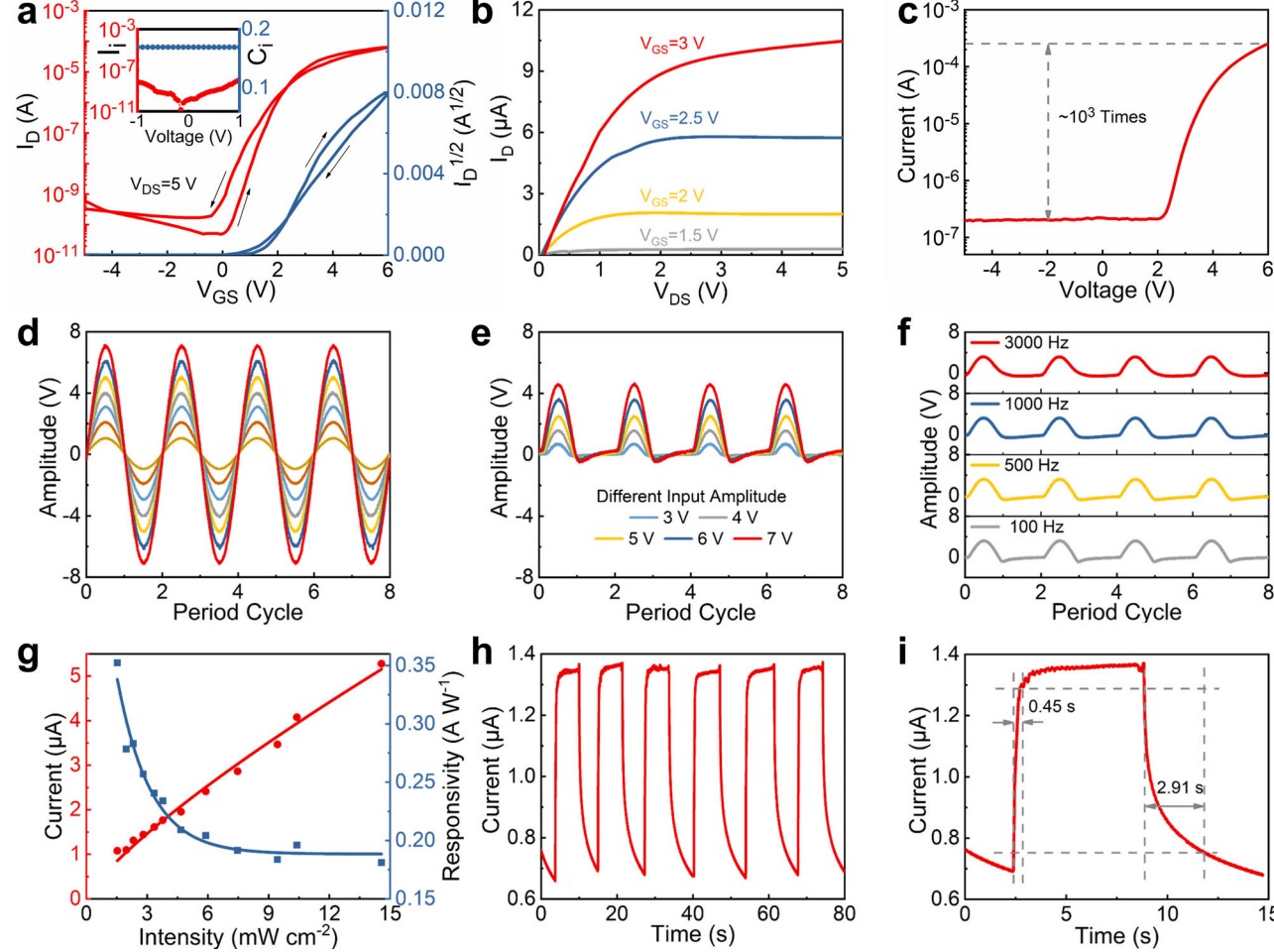

**Fig. 4 | Characterization of the TFT, TFTR and PD at $P_{O2} = 0.04$ Pa. a** Transfer curve, **b** output characteristics, and **c** rectification curve of the TFT device. The inset of Fig. 4a shows areal capacitance ($C_i$, μF cm$^{-2}$) and leakage ($I_i$, A) as a function of applied voltage for the HfLaO-based capacitor. **d** AC sinusoidal signals generated by a signal generator as the TFTR input. **e** Half-wave DC output signals of the TFTR under different-amplitude AC sinusoidal in put signals with a fixed frequency of 100 Hz. **f** Half-wave DC output signals of the rectifier under different-frequency AC sinusoidal input signals with a fixed amplitude of 7 V. **g** Dependence of photocurrent and responsivity on the 405-nm light intensity for the PD device. **h** Current-time curve measured under a 405-nm light intensity of 2.3 mW cm$^{-2}$ at a bias of 5 V for the PD device. **i** High-resolution current-time curve under a 405-nm light intensity of 2.3 mW cm$^{-2}$ at a bias of 5 V used for measuring the rise time and decay time of the PD device.

By shorting the TFT gate to drain terminal, the TFT can work as a rectifier (denoted as TFTR), in which the gate to drain short terminal acts as the rectifier input ($V_{in}$) and the source terminal acts as the rectifier output ($V_{out}$). This rectifier works in two modes: (i) when $V_{in} < V_{th}$, the device is switched-off; (ii) when $V_{in} \geq V_{th}$, the device is turned-on and works in the saturation region. Figure 4c shows the rectification characteristic of the TFT device at $P_{O2} = 0.04$ Pa and a rectification ratio of $1.2 \times 10^3$ can be obtained. In order to check the effectiveness of this rectifier, AC sinusoidal signals with different frequencies and different amplitudes are applied to the rectifier input by using a signal generator (Fig. 4d). The corresponding rectifier output across a load resistance (~50 kΩ) is recorded by using an oscilloscope, as seen in Fig. 4e, f. The output displays half-wave DC signals in all the cases, demonstrating successful rectification. It is further found that the rectifier works well even when the input frequency is up to 3 kHz. Moreover, the rectifier can be also successfully used to process a signal generated by a piezoelectric vibration energy harvester. Supplementary Fig. 13 shows the AC input signal generated by the harvester as well as the corresponding DC output signal processed by the rectifier. This result implies the potential of this TFTR rectifier in constructing a self-powered and autonomous microsystem.

## PD with IGZO as the photosensitive layer

As a typical kind of sensors, PD is used in diverse applications such as flame detection, pollution monitoring, and medical care[37]. In this work, a transparent PD is prepared as the sensing component in this integrated microsystem. This PD presents a two-terminal photoresistive structure and consists of ITO electrode/IGZO photosensitive layer/ITO electrode (Fig. 1a), in which IGZO has been demonstrated to be a promising photosensitive material due to its suitable bandgap and good electrical properties[14,38]. The effects of IGZO prepared at various $P_{O2}$ on the PD performance are investigated. Supplementary Fig. 14 shows the PD current-voltage (I-V) curves under different power intensities at 405-nm light illumination and in the dark. The light-to-dark current ratio (defined as the ratio between the photocurrent at a light intensity of 14.6 mW cm$^{-2}$ and the dark current when the bias is 5 V) is calculated to be 2.9, 48.3, and 728.1 for the PDs prepared at $P_{O2} = 0$, 0.04 and 0.15 Pa, respectively. Owing to its largest dark current resulting from its highest carrier density, the PD at $P_{O2} = 0$ Pa presents the worst light-to-dark current ratio among the devices. In order to further evaluate the photoresponse performance, the relationship between the photocurrent $I_{photo}$ ($I_{photo} = I_p - I_{dark}$. $I_p$ represents the photocurrent under light illumination and $I_{dark}$ means the current in the dark) and the light intensity (P) is also examined and depicted in Fig. 4g. The $I_{photo}$-P relationship can be well described by the following power law[39]

$$I_{photo} = AP^{\theta} \tag{1}$$

where $A$ is a constant and $\theta$ is the empirical value. For the PDs prepared at $P_{O2} = 0$ Pa, 0.04 Pa and 0.15 Pa, the $\theta$ value is estimated to be 0.53, 0.79 and 0.86 by fitting, respectively, revealing the sublinear $I_{photo}$-P relationship. This commonly appears in the metal-oxide-based photodetectors due to the complex processes of electron-hole generation, trapping and recombination in the metal-oxide semiconductors[39]. As a PD critical parameter, responsivity (R) can be calculated by the following equation

$$R = I_{photo}/(PS) \tag{2}$$

where $S$ is the light illumination area. As seen in Fig. 4g and Supplementary Fig. 15, the responsivity decreases and then tends to saturate with increasing the light intensity for all the devices. A maximum responsivity of 6.00, 0.35, and 0.01 A W$^{-1}$ is obtained at a light intensity of 1.5 mW cm$^{-2}$ and a bias of 5 V for the PDs at $P_{O2} = 0$, 0.04 and 0.15 Pa,

respectively. The PD at $P_{O2} = 0$ Pa has the smallest resistance and thus largest photocurrent among the devices, thus leading to its highest responsivity. Based on the above analysis, the PD at $P_{O2} = 0.04$ Pa displays better overall performance in terms of its relatively larger light-to-dark current ratio (vs. the one at $P_{O2} = 0$ Pa) and higher responsivity (vs. the one at $P_{O2} = 0.15$ Pa).

Supplementary Fig. 16 shows the I-V curves under different wavelengths at a power intensity of 7.5 mW cm$^{-2}$ and in the dark for the PD at $P_{O2} = 0.04$ Pa. The current increases with decreasing the light wavelength. An interesting phenomenon is that the PD exhibits a weak photoresponse at a wavelength of 660 nm (the corresponding photon energy is 1.9 eV). This should be ascribed to the oxygen vacancy in IGZO, which would create subgap states in the IGZO bandgap[40]. These subgap states are located at about 1.7 eV below the conduction band minimum given that the IGZO bandgap is 3.2 eV and thus can make a photoresponse at 660-nm light illumination[41,42].

Reproducibility and response speed are another two important parameters of the photodetectors. As shown in Fig. 4h, the PD device at $P_{O2} = 0.04$ Pa presents stable on- and off-state currents under repeatedly chopping the light illumination, confirming its good reproducibility and stability. A high-resolution current-time curve is used to investigate the response speed of the PD device (Fig. 4i). The rise time and decay time are defined as the time interval for the photocurrent changing from 10% to 90% and vise visa, respectively. Accordingly, the rise time and decay time are measured to be 0.45 s and 2.91 s. This PD performance is comparable to the values of the PDs with a similar structure reported in the literature[43–45]. As shown in Fig. 4h and i, a persistent photocurrent is observed after illumination. This is mainly because charge trapping occurs in the IGZO defects induced by light illumination. Two main methods are suggested to suppress this persistent photocurrent: (i) passivating the defects in the IGZO film;[46] (ii) applying a gate bias on the device to compensate this persistent photocurrent[47,48].

## Collaborative work of each device

The collaborative capabilities of each device in the microsystem are firstly demonstrated through the thin-film LIB charging process by using the TFTR as the on-chip rectifier. The detailed setup diagram is presented in Fig. 5a, in which external AC sinusoidal signals created by a signal generator are applied to the input terminal of the TFTR, and the output of the TFTR is connected to the cathode terminal of the LIB. In this case, the switch S1 is turned-on by applying a gate voltage of 6 V through a semiconductor parameter analyzer and the switch S2 is turned-off by floating its gate. Figure 5b shows the LIB charging curves under different-frequency sinusoidal signals with constant amplitude of 7 V. The LIB voltage increases and then tends to saturate with charging time, suggesting that the energy is successful charged into the LIB. An interesting phenomenon is that the charging rate increases with increasing the input-signal frequency (9.3 mV s$^{-1}$ at 800 Hz vs. 8.4 mV s$^{-1}$ at 50 Hz). After charging for 200 s, the input signal is disconnected to the TFTR, and then the LIB is in the rest state. It is found that the LIB voltage decreases at first and then tends to maintain constant, indicating low self-discharge of this LIB. As seen in Fig. 5c, the effects of the input-signal amplitude ($V_p$) on the LIB charging process are also investigated and the LIB charging rate increases with increasing $V_p$ (9.0 mV s$^{-1}$ at $V_p = 7$ V vs. 5.6 mV s$^{-1}$ at $V_p = 4$ V).

Once the thin-film LIB is charged, it can be used as the on-chip power source to drive the PD, as seen in Fig. 5d. In this case, the switch S1 is turned-off and the switch S2 is turned-on. The LIB charged by using a 300-Hz 7 V-$V_p$ sinusoidal signal for 200 s is used to power the PD. Figure 5e shows the PD photoresponse under light illumination with different power densities and a fixed wavelength of 405 nm. The relative resistance variation (defined as $(R - R_O)/R_O \times 100\%$, where $R$ represents the PD resistance under light illumination and in the dark, and $R_O$ is the minimum resistance under light

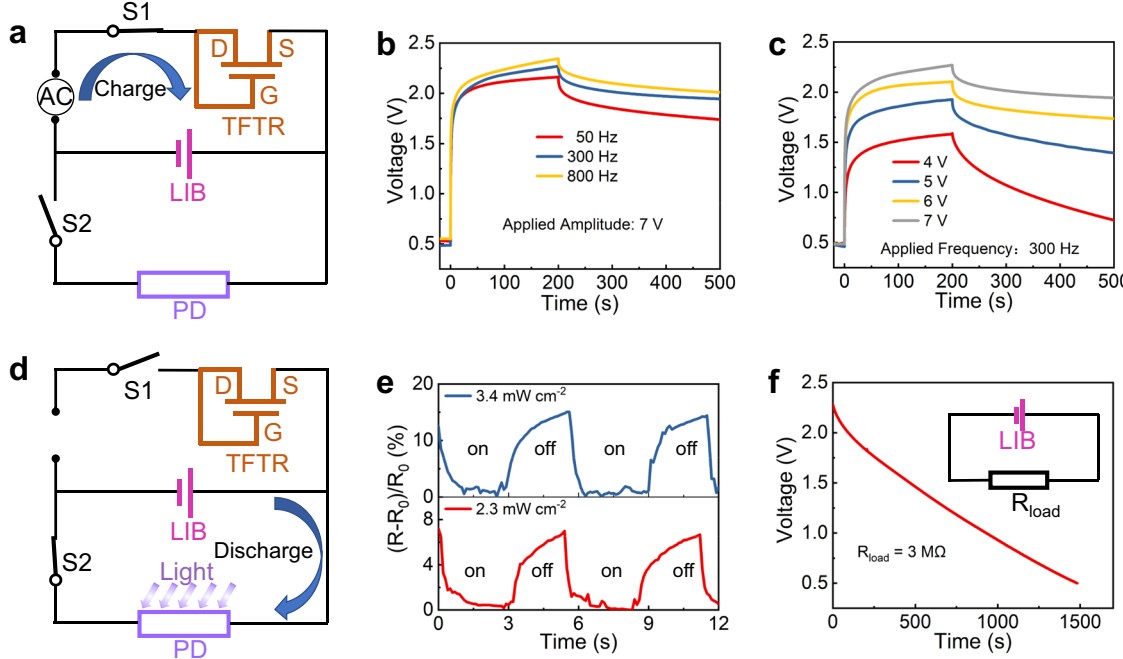

**Fig. 5 | Collaborative operation of each component in the integrated transparent system. a** Test setup of LIB charging process by using TFTR as the on-chip rectifier. In this case, the switch S1 is turned-on and the switch S2 is turned-off. LIB charging process under AC sinusoidal input signals with (**b**) different voltage amplitudes and (**c**) different frequencies. **d** Test setup of PD photoresponse by using the charged LIB as the on-chip power. In this case, the switch S1 is turned-off and the switch S2 is turned-on. **e** PD photoresponse under light illumination with different power densities at a fixed wavelength of 405 nm. **f** Evaluation of operating time for the charged LIB used in the test setup displayed in Fig. 5d.

illumination. -15.1% at 3.4 mW cm$^{-2}$ vs. 6.7% at 2.3 mW cm$^{-2}$) increases obviously with increasing the light power intensity, suggesting high sensitivity of this PD. The circuit diagram in the inset of Fig. 5f is used to estimate the operating time of the charged LIB used in the test setup shown in Fig. 5d, in which the LIB is series connected to an external load resistance $R_{load}$ by using a probe station. $R_{load}$ herein is used to mimic the PD photoresistance and has similar value (~3 MΩ) to the PD photoresistance under light illumination at 405 nm and 3.4 mW cm$^{-2}$. Figure 5f displays the corresponding LIB discharging curve as the LIB voltage decreases to a cut-off voltage of 0.5 V and the operating time is 1482 s in this condition. The above results suggest that the PD can work well by using the LIB as the on-chip power source.

## Discussion

In summary, a transparent thin-film lithium-ion battery with IGZO as the anode is investigated as the on-chip power source. Also, TFT with IGZO as the channel and PD with IGZO the photosensitive layer are also prepared. An integrated and transparent microsystem including energy device (LIB), electronic device (TFT) and sensing device (PD) are successfully developed. Because all the devices use IGZO as their functional layers, this transparent microsystem presents simple fabrication process and compact structure. This study opens up a new avenue for constructing a fully integrated transparent microsystem with function diversity, high integration density and simple fabrication processes.

## Methods

### Device fabrication and optimization

Each device, including LIB, TFT and PD, is individually prepared on glass substrate for investigating and optimizing their performance before fabricating the integrated microsystem. Note that these devices have same dimensions and fabrication processes (the details would be seen later) to their counterparts used in the integrated system. In order to study the effects of the IGZO material properties on each device performance, IGZO films with different oxygen contents were prepared by changing the oxygen partial pressure ($P_{O2}$ = 0 Pa, 0.04 Pa and 0.15 Pa, respectively). Besides glass substrate, IGZO films were also deposited on Si substrate and Cu foil for investigating their material and electrochemical properties. LIB consists of ITO current collector|$V_2O_5$ cathode|LiPON electrolyte|IGZO anode|ITO current collector and its effective area (defined as the overlap between the cathode and anode current collectors) is 3.8 mm × 3.0 mm (width × length). TFT consists of ITO gate/HfLaO dielectric/IGZO channel/ITO source and drain and the channel width and length are 300 μm and 30 μm, respectively. In addition, a capacitor with a metal-insulator-metal (MIM, top electrode/dielectric/bottom electrode ~ Au-Ti/HfLaO/ITO) structure was also prepared for investigating the HfLaO dielectric properties, where the top electrode was fabricated by using electron-beam evaporation (Xingnan Zss) and the dielectric and bottom electrode were prepared by using sputtering as seen in Supplementary Table 1. PD displays a photoresistive structure and consists of ITO electrode/IGZO photosensitive layer/ITO electrode and the resistance width and length are 500 and 150 μm, respectively. LIB and PD were fabricated by using a magnetron sputtering system (Korvus Hex Deposition System). Because TFT is sensitive to the metallic contamination, its ITO layers (including the gate and source/drain) were fabricated by using the Korvus Hex Deposition System and its HfLaO and IGZO layers were deposited by using another magnetron sputtering system (Sky Deposition System), which was only used for dielectric and semiconductor deposition and thus the metallic contamination is effectively suppressed.

### Microsystem fabrication

Based on the experimental results, the devices (including the LIB, TFT and PD) at $P_{O2}$ = 0.04 Pa present better overall performance than the ones at $P_{O2}$ = 0 Pa and 0.15 Pa. Therefore, the IGZO film prepared in this condition was used for constructing the integrated microsystem. It is

also demonstrated that $P_{O_2}$ has a great influence on device performance and thus a careful optimization is worthy to be performed in the future. The fabrication was carried out by using physical vapor deposition (targets purchased from Kurt J. Lesker) and each layer was patterned by using shadow masks and the processes were shown in Supplementary Fig. 1. The detailed deposition conditions of each layer were summarized in Supplementary Table 1. The fabrication was carried out at room temperature to ensure that the active films are in the amorphous state. This helps address the nonuniformity issues of a microsystem caused by random grain boundaries in the films. The fabrication started with an ultrasonic cleaning of the glass substrate (purchased from Kintec, 8 mm × 8 mm) in the acetone, isopropanol and deionized water for 15 min successively. After cleaning, 80-nm ITO film was deposited on the substrate as the LIB current collector and TFT bottom gate, respectively. After that, 230-nm $V_2O_5$ was prepared as the LIB cathode. Then, 660-nm LiPON was sputtered as the LIB electrolyte. Following that, 90-nm HfLaO was deposited as the TFT gate dielectric. Next, 80-nm IGZO film was deposited as the LIB anode, TFT channel and PD photosensitive layer, respectively. Finally, 80-nm ITO was prepared to form the LIB current collector, PD electrode, TFT source/drain electrode and interconnects between each device, respectively.

## Material characterization

The thickness of the films was characterized by using spectroscopic ellipsometry (Horriba UVISEL) and confirmed by scanning electron microscopy (SEM, Helios 5 CX) equipped with energy dispersive X-ray spectroscopy (EDX). The compositions of the samples were studied by using Raman spectroscopy (Horiba HR-800), SEM and EDX. The morphologies of the samples were characterized by using SEM. The crystallinity of the samples was studied by using X-ray diffraction (XRD, Rigaku Smartlab3) and transmission electron microscopy (TEM, FEI-Themis Z). The Hall carrier concentration and mobility of the samples were measured by a Hall-effect measurement system (Ecopia HMS-7000). The mass loading of the films was determined by the difference between the substrate masses before and after film deposition (Mettler Toledo PR504S/AC).

## Device and microsystem characterization

The UV−vis transmittance spectra of the devices and integrated microsystem were recorded by using a spectrophotometer (Yipu Instrument, U-T1810D). For the electrochemical measurements, CR2016-type coin half cells (the diameter ~20 mm) with IGZO film on the Cu foils as the anode (the mass loading ~0.027 mg cm$^{-2}$ for 400-nm IGZO), Celgard 2400 as the separator, 1.0 M LiPF$_6$ in ethylene carbonate (EC) and ethyl methyl carbonate (EMC), (EC/EMC = 50/50 (v/v), from Sigma-Aldrich) as the electrolyte and Li foil as the counter electrode were assembled and prepared. The electrochemical characterization of the half cells and all-solid-state thin-film LIBs was performed at an Ar-filled glove box (Mikrouna). The cyclic voltammetry (CV) was recorded by using an electrochemical workstation (CHI 660E). The galvanostatic charge/discharge (GCD) measurements at various current densities were carried out by using a battery test system (Lanhe M340A). The electrochemical impedance spectra (EIS) were measured over the frequency range from $10^5$ to $10^{-1}$ Hz by applying an AC voltage with 5-mV amplitude (CHI 660E). The capacitance and leakage of the HfLaO-based capacitor were measured by using an impedance analyzer at 1 MHz (Agilent 4294 A) and a semiconductor parameter analyzer (Keithley 4200-SCS), respectively. The electrical performance of the TFT and PD devices as well as the microsystem was measured mainly based on a semiconductor parameter analyzer (Keithley 4200-SCS) together with an ARS cryogenic probe station. The measurements were performed at a vacuum pressure of $10^{-3}$ mbar to suppress the influences of the atmosphere on the device performance. For the rectifying measurements, sinusoidal signals created by a signal generator (Tektronix AFG1000) were used as the input. Besides, AC signals generated by repeatedly pressing a homemade piezoelectric energy harvester were also used as the input. The input and output signals were recorded by using an oscilloscope (Rigol DS1102E). For the PD measurements, laser lamps with wavelengths 405 nm, 450 nm and 660 nm (WYoptics) were used as the light sources. All the measurements were conducted at 25 °C unless specifically stated.

## Data availability

All the data that support the findings of this study are available within the main text and Supplementary Information file, and also available from the corresponding author on request. Source data are provided with this paper.

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

## Acknowledgements

This work was supported in part by the National Key R&D Program of China under Grants No. 2020YFB2007400 (X.D.H.) and in part by the National Natural Science Foundation of China under Grants No. 61974026 (X.D.H.) and No. 62174130 (C.Y.H.).

## Author contributions

X.H. planned and supervised the research. B.J. performed the data collection and drafted the manuscript. C.Z. and J.W. helped to analysis the data. Z.L. helped the layout and optimization of pictures. M.L. and L.Z. performed the microscope characterizations. C.H. and M.Q. help designed and conducted the experimental work. X.H. proofread the manuscript. All authors participated in discussing the results and finalizing the manuscript.

## Competing interests

The authors declare no competing interests.
