## [Peer Review File · Nature Communications]

Integration of microbattery with thin-film electronics for constructing an integrated transparent microsystem based on InGaZnOREVIEWER COMMENTS

Reviewer #1 (Remarks to the Author):

InGaZnO (IGZO) is an important semiconductor. IGZO TFT and photodetector have been widely studied.

However, there is still lack of transparent thin-film lithium-ion battery (LIB) with IGZO as the anode, Although InOx, GaOx, ZnOx have been used in LIB.

This work reported a transparent LIB with IGZO as the anode, TFT with IGZO as the channel layer and PD with IGZO as the photosensitive layer are prepared on a single glass substrate for constructing an integrated transparent microsystem.

It would be of significance to the field of transparent electronics.

IGZO thin films require the performance characterization, such as Hall effect, showing the electronics data from different oxygen partial pressures.

“Based on the above results, The IGZO film prepared at an oxygen partial pressure of 0.04 Pa shows better overall electrochemical performance than the other ones in terms of its higher reversible capacity, longer cycling life and better rate characteristics. Therefore, the film prepared in this condition is used as the functional layers in each essential device and the integrated microsystem in the following.”

The detailed requirement of IGZO for different devices, LIB, TFT and PD may be different. Thus the data of these 3 devices based on IGZO may need further optimization.

Each device displays an acceptable performance. However, In Fig. 5, on/off current ratio ($I_{on} / I_{off} \sim 103$) is too low for TFT. What is the reason? Could TFT device be improved further?

In particular, the requirement of electrochemical and semiconductor performance may be significantly different for IGZO.

The performances of devices are important to the related fields.

The properties of LIB and PD should be optimized individually, because each device may require the different properties of IGZO layers.

Reviewer #2 (Remarks to the Author):

The manuscript reported the development of integration of IGZO-based lithium-ion battery (LIB), thin-film transistor (TFT) and photodetector (PD) as the transparent microsystem. A simple fabrication process and compact structure was demonstrated to prepare for such microsystem. One of the highlights of this work is the application of IGZO as the anode to form a transparent LIB. The details about the fabrication processes, performance of individual device, and the results of collaborative work of each device had been well described to guaranty the feasibility of this proposed system.

Despite this, deeper purpose analyses of this proposed transparent system are anticipated in order to make this work more understandable and agreeable. It is thus advised to add more reviews and descriptions of relevant research reports in the introduction which is quite simple but lacking in information. For example, the need to integrate all transparent energy device, electronic device and sensing device in one substrate. What is the required scope of this transparent system on the application side? What are the reasons for the integration of LIB, TFT and PD components and the possibility of its application? Especially, what is the purpose of developing transparent LIB?

For the application of IGZO as the anode to form a transparent LIB, it is essential to consider the theoretical predictions on the performance (such as theoretical capacity (mA h g^{-1})) of IGZO metal oxide as electrodes for metal ion batteries, together with the discussion on the difference between experimental results and theoretical analyses.

Though the oxygen partial-pressure dependence of electrochemical performance of IGZO film had been studied in this work, comparisons with more recent reports of other metal oxides/composites

in terms of the reversible capacity, cycling life, cycling stability and rate characteristics are of reference value for academic research and practical application.

Reviewer #3 (Remarks to the Author):

The authors describe the fabrication and characterization of transparent IGZO based electronic devices (thin-film transistor (TFT), photodetector (PD) and lithium ion battery (LIB)) within a simultaneous processing route on a single glass substrate. While especially IGZO TFTs and PDs are not themselves new, the combination of all transparent elements into a functional and self-powered monolithic unit is an advancement for potential future applications in the realm of Internet of things and consumer electronics. Given the broad scope it is expected that a wider audience would be interested in the current findings, however, there are still a number of open questions that should be resolved:

1. Within the individual elements of the presented microsystem, the transparent thin-film battery stands out with the highest amount of novelty and innovation. As such, it is suggested to add a little more background information to bring it into context with state-of-the-art literature. In addition, the references 33-35, given as comparison for V2O5-based thin-film LIBs are from 2001 to 2004. How about more recent publications?
2. The illustration of the microsystem in Fig. 1a shows multiple discrete TFTs and PDs on the substrate whereas for the complete system more interconnects are required between the respective parts. How were such interconnects realized, through external wiring or with a separate deposition of e.g. ITO electrodes?
3. While the process parameters for each respective layers are well documented in the Supporting Information, neither there or in the main manuscript can information be found on the actual device dimensions that were used in the study (e.g. TFT channel length/width, etc). This information needs to be included for all components as well as the substrate.
4. Some questions regarding the TFTs:
 - i. Did the devices show any hysteresis? Please include forward and backward sweep in 5a)
 - ii. What was the capacitance of the dielectric and at what frequency was it measured at?
 - iii. The off-current at negative V_g is rather high. What is believed to be the limiting factor here? Is it related to the IGZO channel material or does the dielectric play a role? Including the gate current in Fig. 5a) could help to clarify this.
 - iv. Although a good Ohmic contact between IGZO and ITO was remarked on by the authors, there still seems to be a noticeable S-shape for V_{ds} values close to 0 V, that is typically indicative of contact resistance. A comment on this behavior would be appreciated.
3. It is somewhat surprising to see an IGZO photoresponse at a wavelength of 660 nm. Could the authors please elaborate on this part? In other studies (compare e.g. Kim et al., J Mater Chem, 8, 165-172, 2020, <https://doi.org/10.1039/C9TC04982G>) defects that create subgap states are specifically introduced into IGZO to make this possible.
4. Metal oxide semiconductors can suffer from a persistent photocurrent after illumination (compare e.g. Jeon et al., Nat. Mater., 11, 301-305, 2012, <https://doi.org/10.1038/nmat3256>). Was any special precaution taken or procedure carried out to prevent this?
5. The transparency of the full microsystem is given as larger than 76 % at a wavelength of over 700 nm. This is borderline in the visible spectrum and for a representative quantification it is recommended to either pick a more central wavelength (e.g. 550 nm) or to compute an average over the visible spectrum.
6. The whole fabrication scheme was kept to room temperature depositions with the aim to keep the active films amorphous and thus to reduce nonuniformities arising from grain boundaries. Adding additional device data related to uniformity and/or stability would strengthen this approach.
7. What was the measurement environment? Since a cryogenic chamber was mentioned, were the tests carried out under vacuum?
8. Are there any estimations for how long the microsystem could be operated on a single battery charge?

Reviewer #4 (Remarks to the Author):

The manuscript entitled "Integration of microbattery with thin-film electronics for constructing an integrated and transparent microsystem based on InGaZnO", by Bin Jia, et al. presents transparent IGZO based batteries, TFTs, and a light sensor on a glass substrate. The paper is well written and shows the very relevant and important realization of a transparent thin-film power source for wearable systems. At the same time, the manuscript is a bit unclear about the integration of all components (based in the current version I assume they are actually not integrated into a system). Hence, I see the following options:

- Focus on the battery part - which is novel, and the related material/device characterization. Then submit this work to a more specialized journal.
- If it is true, that the individual components were not connected on the substrate I would not consider this a microsystem. This would make it difficult to publish the work in NatCom
- If there is a fully integrated system, or if there will be one after some revision, I would support accepting the work, if also the following comments are addressed.

Additional more technical comments are:

1. Fig 1e quantifies the transmittance, but in addition I would expect to see a much higher resolution and clearer photograph of the system.
2. The novelty of Fig. 2 compared to the state of the art in the context of IGZO is very limited. However, it could be part of the supplementary information.
3. The craterisation of the battery is not fully consistent. E.g. it would be better to show a full set of numbers and graphs for the same current densities.
4. Transistors:
 - The achieved performance needs more discussion, in particular the current ratio is not necessarily in line with the state of the art
 - The text mentions good ohmic contacts, but Fig 5b actually looks like there is significant impact of the contact resistance at low V_{ds} (although it is hard to say as the resolution is a bit low)
 - Fig.5a needs more details such as the W/L ratio, the gate leakage current, the current in the linear region, etc..
 - The manuscript mentions the turn-on voltage and then treats it like the threshold voltage, however these are not the same. Please clarify this.
 - Figs 5d and 5e should use the same scale, and fig 5f should also show a scale (even if there is a vertical offset used, the difference in voltage for all measurements should be shown)
5. Photo detectors:
 - Effectively this device is a resistive sensor not a photodiode so it does not make a lot of sense to show the current (as it depends on the applied voltage – which is not given for all measurements). Here the resistance variation of the device should be shown.
6. Please give the dimensions of all devices.
7. Integration:
 - Specify how the devices were connected in Fig 5.
 - How were the switches S1 and S2 implemented?
 - Please show a micrograph of the full system.
 - It is written that after charging the battery the AC input signal was "shut down". Please specify this, was the input 0V or floating? Was it still connected to the TFT? This would most probably influence the discharge.

Point-by-point response to the reviewers' comments

Reviewer #1

InGaZnO (IGZO) is an important semiconductor. IGZO TFT and photodetector have been widely studied.

However, there is still lack of transparent thin-film lithium-ion battery (LIB) with IGZO as the anode, Although InO_x , GaO_x , ZnO_x have been used in LIB.

This work reported a transparent LIB with IGZO as the anode, TFT with IGZO as the channel layer and PD with IGZO as the photosensitive layer are prepared on a single glass substrate for constructing an integrated transparent microsystem.

It would be of significance to the field of transparent electronics.

Response: We really appreciate the reviewer's positive comments on the quality of this manuscript.

1. IGZO thin films require the performance characterization, such as Hall effect, showing the electronics data from different oxygen partial pressures.

Response: As seen in Supplementary Fig. 2c, the Hall carrier concentration and mobility of the IGZO films prepared at different oxygen partial pressures (P_{O_2}) have been measured by using a Hall-effect measurement system (Ecopia HMS-7000). It can be found that the Hall carrier concentration and mobility decrease with increasing P_{O_2} . Oxygen vacancy (V_O) is the main contributor to the electron carriers and electrical conductivity of IGZO (*J. Felizco, et al., Appl. Surf. Sci., 527, 146791, 2020.*). Increasing P_{O_2} enhances the oxygen content (and thus suppresses the V_O content) in IGZO, thus reducing the Hall carrier concentration and mobility.

The revisions have been added on L. 104-111, P. 5; L. 469-471, P. 20 in the manuscript; Supplementary Fig. 2c, P. 3 in the Supplementary Information.

Supplementary Fig. 2. Material characterization of the IGZO films prepared at different P_{O_2} . **(a)** Raman spectra. **(b)** EDX spectra. **(c)** Hall carrier concentration and mobility. **(d)** XRD patterns.

2. Based on the above results, The IGZO film prepared at an oxygen partial pressure of 0.04 Pa shows better overall electrochemical performance than the other ones in terms of its higher reversible capacity, longer cycling life and better rate characteristics. Therefore, the film prepared in this condition is used as the functional layers in each essential device and the integrated microsystem in the following.”

The detailed requirement of IGZO for different devices, LIB, TFT and PD may be different.

Thus the data of these 3 devices based on IGZO may need further optimization.

Response: As the reviewer mentioned, different devices (including LIB, TFT and PD) may have different demands on IGZO. The effects of IGZO prepared at various P_{O_2} on the LIB performance were carefully investigated in our previous manuscript, which shows that the IGZO anode at $P_{O_2} = 0.04$ Pa displays better electrochemical performance than the ones at $P_{O_2} = 0$ Pa and $P_{O_2} = 0.15$ Pa in terms of its higher

reversible capacity, better cycling stability and better rate characteristics, as seen in Supplementary Table 4. This is mainly because of its higher ionic conductivity (vs. the anode at $P_{O_2} = 0$ Pa) and higher electrical conductivity (vs. the anode at $P_{O_2} = 0.15$ Pa).

Supplementary Table 4. Performance comparison of the devices (including LIB, TFT and PD) prepared at various P_{O_2} .

P_{O₂}		0 Pa	0.04 Pa	0.15 Pa
LIB	Specific capacity (mAh g ⁻¹)	845.8	989.6	942.8
	Capacity retention (%)	67.1%	91.5%	94.8%
	Rate performance (mAh g ⁻¹)	17 (4 Ag ⁻¹) 274 (0.2A g ⁻¹)	227 (4 Ag ⁻¹) 616 (0.2A g ⁻¹)	149 (4 Ag ⁻¹) 310 (0.2A g ⁻¹)
TFT	Mobility (cm ² V ⁻¹ s ⁻¹)	58.0	23.3	8.3
	Sub-threshold swing (mV dec ⁻¹)	458.9	209.2	180.0
	Threshold Voltage (V)	1.8	1.9	2.3
	Hysteresis (V)	0.6	0.4	0.3
	On-off current ratio	5.1×10^5	1.4×10^6	4.2×10^5
PD	Responsivity (A W ⁻¹)	6.0	0.35	0.01
	Light-to-dark current ratio	2.9	48.3	728.1

In the revised manuscript, the effects of IGZO prepared at various P_{O_2} on the TFT and PD performance have also been investigated respectively, as seen in in Fig. R1 and

R2. The critical parameters of the devices are also extracted from these figures and summarized in Supplementary Table 4 above. For the TFT devices, the TFT at $P_{O_2} = 0.04$ Pa displays higher carrier mobility ($23.3 \text{ cm}^2 \text{ V}^{-1} \text{ s}^{-1}$ vs. $8.3 \text{ cm}^2 \text{ V}^{-1} \text{ s}^{-1}$) and larger I_{on}/I_{off} ratio (1.4×10^6 vs. 4.2×10^5) than that at $P_{O_2} = 0.15$ Pa mainly because of its more V_O content and thus higher carrier density. However, excessive V_O content results in excessive traps in IGZO, which causes that the TFT at $P_{O_2} = 0$ Pa presents much larger hysteresis (0.6 V vs. 0.4 V) and larger sub-threshold swing ($458.9 \text{ mV dec}^{-1}$ vs. $209.2 \text{ mV dec}^{-1}$) than the one at $P_{O_2} = 0.04$ Pa. For the PD devices, the light-to-dark current ratio and responsivity decrease and increase respectively with increasing P_{O_2} . The PD at $P_{O_2} = 0.04$ Pa has much smaller dark current (10^{-7} A vs. 10^{-5} A at a bias of 5 V) than the one at $P_{O_2} = 0$ Pa mainly because of its lower carrier density, thus resulting in its higher light-to-dark current ratio. In addition, the PD at $P_{O_2} = 0.04$ Pa has smaller resistance and thus larger photocurrent (10^{-6} A vs. 10^{-7} A under a 405-nm light intensity of 14.6 mW cm^{-2} at a bias of 5 V) than the one at $P_{O_2} = 0.15$ Pa under the same operating conditions, thus leading to its better responsivity (0.35 A W^{-1} vs. 0.01 A W^{-1}) than the latter.

Therefore, based on the above discussions, the device (including LIB, TFT and PD) at $P_{O_2} = 0.04$ Pa displays better overall performance than the ones at $P_{O_2} = 0$ Pa and 0.15 Pa. Therefore, the IGZO film prepared in this condition is used for constructing the integrated system.

Fig. R1. Transfer curves of the TFTs with IGZO channel layer prepared at (a) $P_{O_2} = 0$ Pa, (b) $P_{O_2} = 0.04$ Pa and (c) $P_{O_2} = 0.15$ Pa. The V_{th} is obtained from the intercept of $I_D^{1/2}$ vs. V_G in the saturation region with the x -axis. ΔV_{th} is defined by the threshold-voltage difference of the transfer curves under forward and backward sweepings.

Fig. R2. $I-V$ curves of the PD devices under different power intensities at 405-nm light illumination and in the dark. (a) PD at $P_{O_2} = 0$ Pa, (b) PD at $P_{O_2} = 0.04$ Pa and (c) PD at $P_{O_2} = 0.15$ Pa. Dependence of photocurrent and responsivity on the 405-nm light intensity for the PD devices at (d) $P_{O_2} = 0$ Pa, (e) $P_{O_2} = 0.04$ Pa and (f) $P_{O_2} = 0.15$ Pa.

The revisions have been added on L. 249-160, P. 11; L. 300-305, P. 13; L. 306-307, 313-314, 320-328, P. 14; Fig. 4, P. 12 in the manuscript; Supplementary Fig. 13; Supplementary Fig. 15, P. 16; Supplementary Fig. 16, P. 17; Supplementary Table 4, P. 22 in the Supplementary Information.

3. Each device displays an acceptable performance. However, In Fig. 5, on/off current ratio ($I_{on}/I_{off} \sim 10^3$) is too low for TFT. What is the reason? Could TFT device be improved further?

Response: The low on/off current ratio ($I_{on}/I_{off} \sim 10^3$) should be mainly ascribed to metallic contamination induced during depositing the TFT films since the transistor is quite sensitive to the metallic contamination. In order to validate this analysis, we have re-prepared the TFT device based on the same fabrication processes. The main difference between the re-prepared TFT and previous TFT is that the dielectric and IGZO films are deposited by using a ‘clean’ sputter, which is only used for dielectric

and semiconductor deposition (not allowed for metal deposition) and thus the metallic contamination can be effectively suppressed. As shown in Fig. R3 below, the re-prepared TFT displays better performance than the previous one in terms of its higher I_{on}/I_{off} ratio (1.4×10^6 vs. 3.4×10^3), higher carrier mobility ($23.3 \text{ cm}^2 \text{ V}^{-1} \text{ s}^{-1}$ vs. $4.5 \text{ cm}^2 \text{ V}^{-1} \text{ s}^{-1}$) and smaller sub-threshold swing ($209.2 \text{ mV dec}^{-1}$ vs. $434.3 \text{ mV dec}^{-1}$). It is noted that the main contribution of this manuscript is to develop an IGZO-based transparent LIB and integrated transparent microsystem. The above results also suggest that it has potential to further improve the performance of this integrated microsystem by optimizing the fabrication conditions.

The revisions have been added on L. 430-435, P.18; Fig. 4a-c, P. 12 in the manuscript; Supplementary Table 1, P. 19 in the Supplementary Information.

Fig. R3. Transfer curves of the TFTs (a) in the current work and (b) in the previous work.

4. In particular, the requirement of electrochemical and semiconductor performance may be significantly different for IGZO.

The performances of devices are important to the related fields.

The properties of LIB and PD should be optimized individually, because each device may require the different properties of IGZO layers.

Response: According to the reviewer's suggestion, the performance of the LIB, TFT and PD devices is optimized individually by changing P_{O_2} during the IGZO deposition.

The detailed results are summarized in Supplementary Table 4 above. It can be found that the LIB, TFT and PD devices prepared at $P_{O_2} = 0.04$ Pa present better overall performance than those at $P_{O_2} = 0$ Pa and 0.15 Pa. Therefore, the IGZO film prepared at $P_{O_2} = 0.04$ Pa is used in the integrated system.

The revisions have been added on L. 249-160, P. 11; L. 300-305, P. 13; L. 306-307, 313-314, 320-328, P. 14; Fig. 4, P. 12 in the manuscript; Supplementary Fig. 13; Supplementary Fig. 15, P. 16; Supplementary Fig. 16, P. 17; Supplementary Table 4, P. 22 in the Supplementary Information.

Reviewer #2

The manuscript reported the development of integration of IGZO-based lithium-ion battery (LIB), thin-film transistor (TFT) and photodetector (PD) as the transparent microsystem. A simple fabrication process and compact structure was demonstrated to prepare for such microsystem. One of the highlights of this work is the application of IGZO as the anode to form a transparent LIB. The details about the fabrication processes, performance of individual device, and the results of collaborative work of each device had been well described to guaranty the feasibility of this proposed system.

Response: We really appreciate the reviewer's positive comments on the quality of this manuscript.

1. Despite this, deeper purpose analyses of this proposed transparent system are anticipated in order to make this work more understandable and agreeable. It is thus advised to add more reviews and descriptions of relevant research reports in the introduction which is quite simple but lacking in information. For example, the need to integrate all transparent energy device, electronic device and sensing device in one substrate. What is the required scope of this transparent system on the application side? What are the reasons for the integration of LIB, TFT and PD components and the possibility of its application? Especially, what is the purpose of developing transparent LIB?

Response: Transparent electronics have promising applications in the next-generation consumer electronics (e.g. transparent displays, transparent mobile phones and laptops), smart home and transportation (e.g. transparent solar cells and smart windows), and securities (e.g. invisible cameras and detectors). Enhancing the functions is one of the main trends driving the development of transparent electronics. This can be achieved by integrating other kinds of components (e.g. sensors and LIBs) with electronic devices (e.g. TFTs) for constructing a microsystem, which is also known as the More than Moore strategy (*H. Faber. et al., Nat. Electron., 2, 497-498, 2019.*). In this

microsystem, LIB works as the power source for other components (e.g. sensors and TFTs). Particularly, for portable transparent electronics (e.g. transparent mobile phones and wireless detectors), it is difficult to use electrical wirings for power supply and thus LIB is essential in these applications. Integration of various components in one substrate (called as monolithic integration) can make the microsystem more compact and lightweight (*W. Q. Wei et al., Light Sci. Appl., 12, 84 2023.*). Moreover, monolithic integration is effective to improve working speed and suppress power consumption by reducing the connection length and relevant parasitic effects between each component. Therefore, a transparent LIB with IGZO as the anode is proposed as the on-chip power source in this work. Then, a monolithic integrated microsystem including the LIB, TFT and sensor devices is developed based on IGZO as their functional layers.

The revisions have been added on L. 29-43, P. 2 in the manuscript.

2. For the application of IGZO as the anode to form a transparent LIB, it is essential to consider the theoretically predictions on the performance (such as theoretical capacity (mA h g^{-1})) of IGZO metal oxide as electrodes for metal ion batteries, together with the discussion on the difference between experimental results and theoretical analyses.

Response: According to the literature (*W.H. Ho, et al., J. Power Sources, 897, 175, 2008; J. Guo, et al., ACS Sustainable Chem. Eng., 13692, 8, 2020; Y.Q. Cao, et al., Sci. Rep., 11526, 9, 2019.*), the theoretical capacity C_g (mAh g^{-1}) of the IGZO compositions (including In_2O_3 , Ga_2O_3 and ZnO) can be calculated based on the following equations

$$C_g = \frac{F \times n}{3.6 \times W} \quad (3)$$

where F is the Faraday constant (96485 C mol^{-1}), W is the molecular weight and is equal to 277.6, 187.4 and 81.0 for In_2O_3 , Ga_2O_3 and ZnO respectively, and n is the number of transferred electrons and can be calculated by

$$n = 2y + x \times z \quad (4)$$

n is determined to be 14.66, 10 and 3 for In_2O_3 , Ga_2O_3 and ZnO , respectively.

Based on the above equations, the theoretic capacity for In_2O_3 ($C_{\text{In}_2\text{O}_3}$), Ga_2O_3 ($C_{\text{Ga}_2\text{O}_3}$) and ZnO (C_{ZnO}) is calculated to be 1415 mAh g^{-1} , 1430 mAh g^{-1} and 987 mAh g^{-1} respectively, which are consistent with those reported in the literature (*W.H. Ho, et al., J. Power Sources, 897, 175, 2008; J. Guo, et al., ACS Sustainable Chem. Eng., 13692, 8, 2020; Y.Q. Cao, et al., Sci. Rep., 11526, 9, 2019.*).

After that, the IGZO theoretic capacity (C_{IGZO}) can be calculated by

$$C_{\text{IGZO}} = C_{\text{In}_2\text{O}_3} \times m_{\text{In}_2\text{O}_3} + C_{\text{Ga}_2\text{O}_3} \times m_{\text{Ga}_2\text{O}_3} + C_{\text{ZnO}} \times m_{\text{ZnO}} \quad (5)$$

where $m_{\text{In}_2\text{O}_3}$, $m_{\text{Ga}_2\text{O}_3}$ and m_{ZnO} represent mass ratio of In_2O_3 , Ga_2O_3 and ZnO in the IGZO film (InGaZnO_4 according to the target used in this work), and are equal to 0.443, 0.299 and 0.258, respectively. Consequently, C_{IGZO} is determined to be 1311 mAh g^{-1} . The experimental capacity (\sim about 1050 mAh g^{-1} at 0.05 A g^{-1}) is a little smaller than the theoretic value. The electrochemical reactions displayed in equations (1) and (2) enhance by decreasing the charge/discharge current, thus helping make the experimental capacity closer to the theoretic value.

The revisions have been added on L. 140-146, P. 7 in the manuscript; Supplementary Note 1, P. 25 in the Supplementary Information.

3. Though the oxygen partial-pressure dependence of electrochemical performance of IGZO film had been studied in this work, comparisons with more recent reports of other metal oxides/composites in terms of the reversible capacity, cycling life, cycling stability and rate characteristics are of reference value for academic research and practical application.

Response: As seen in Supplementary Table 5, a comparison of the IGZO anode film in this work with other typical metal-oxide/composite anode films in the literature has been performed. The IGZO anode film displays desirable electrochemical performance based on this comparison.

The revisions have been added on L. 180-182, P. 8 in the manuscript; Supplementary Table 5 in the Supplementary Information.

Supplementary Table 5. Performance comparison of the IGZO anode film at $P_{O_2} = 0.04$ Pa in this work with other typical metal-oxide anode films in the literature.

Anode materials	Reversible capacity (mAh g ⁻¹)	Initial CE (%)	Cycle number (capacity retention)	Rate performance (mAh g ⁻¹)	Ref.
In ₂ O ₃	195 @50μA cm ⁻²	10.0	10 (22.4%)	/	1
ZnO	435 @0.2A g ⁻¹	56.1	100 (40.0%)	435 @0.2 A g ⁻¹ ; 175 @3.2 A g ⁻¹	2
Al ₂ O ₃ /ZnO	301 @125 mA g ⁻¹	/	200 (49.8%)	/	3
TiO ₂	327 @C/3	/	200 (77.6%)	/	4
TiO ₂ /SiO ₂	560 @50μA cm ⁻²	56.6	100 (76.8%)	560 @50 μA cm ⁻² ; 242 @1000 μA cm ⁻²	5
CuO/TiO ₂	1036 @0.1C	60.5	10 (62.6%)	1036 @0.1C; 167 @2.0C	6
Cu ₂ O/TiO ₂	520 @0.1C	53.6	10 (89.5%)	520 @0.1C; 370 @2.0C	6
Fe ₂ O ₃	951 @100mA g ⁻¹	71.0	200 (77.2%)	951 @100 mA g ⁻¹ ; 510 @15 A g ⁻¹	7

Fe ₂ O ₃ /Ag	1167 @0.1A g ⁻¹	72.5	50 (83.5%)	1167 @0.1C; 509 @20.0C	8
LiPON/ NiFe ₂ O ₄	917 @5μA cm ⁻²	67.7	50 (65.3%)	917 @5 μA cm ⁻² ; 650 @25 μA cm ⁻²	9
NiO	1134 @ 0.1A g ⁻¹	73.6	50 (45.5%)	1134 @0.1 A g ⁻¹ ; 382 @0.8 A g ⁻¹	10
InGaZnO	990 @50 mA g ⁻¹	70.0	250 (89.2%)	616 @200 mA g ⁻¹ ; 227 @4 A g ⁻¹	This work

References

1. Ho, W. H., Li, C. F., Liu, H. C. & Yen, S. K. Electrochemical performance of In₂O₃ thin film electrode in lithium cell. *J. Power Sources* **175**, 897-902 (2008).
2. Yuan, J. et al. Facile fabrication of three-dimensional porous ZnO thin films on Ni foams for lithium ion battery anodes. *Mater. Lett.* **190**, 37-39 (2017).
3. Shi, Q. et al. Electrochemical and optoelectric behavior of Al-doped ZnO films as transparent anode for Li-ion batteries. *Mater. Today Commun.* **19**, 471-475 (2019).
4. Nagpure, S. et al. Layer-by-layer synthesis of thick mesoporous TiO₂ films with vertically oriented accessible nanopores and their application for lithium-ion battery negative electrodes. *Adv. Funct. Mater.* **28**, 1801849 (2018).
5. Wu, J. et al. Rapid construction of TiO₂/SiO₂ composite film on Ti foil as lithium-ion battery anode by plasma discharge in solution. *Appl. Phys. Lett.* **114**, 043903 (2019).
6. Barreca, D. et al. On the performances of Cu_xO-TiO₂ (x = 1, 2) nanomaterials as innovative anodes for thin film lithium batteries. *ACS Appl. Mater. Interfaces* **4**, 3610-3619 (2012).
7. Teng, X. et al. A nanocrystalline Fe₂O₃ film anode prepared by pulsed laser deposition for lithium-ion batteries. *Nanoscale Res. Lett.* **13**, 60 (2018).
8. Zhang, D., Li, Y., Yan, M. & Jiang, Y. Fe₂O₃-Ag porous film anodes for ultrahigh-

rate lithium-ion batteries. *ChemElectroChem* **1**, 1155-1160 (2014).

9. Wei, K. et al. Lithium phosphorous oxynitride (LiPON) coated NiFe₂O₄ anode material with enhanced electrochemical performance for lithium ion batteries. *J. Alloys Compd.* **769**, 110-119 (2018).

10. Cao, L., Wang, D. & Wang, R. NiO thin films grown directly on Cu foils by pulsed laser deposition as anode materials for lithium ion batteries. *Mater. Lett.* **132**, 357-360 (2014).

Reviewer #3

The authors describe the fabrication and characterization of transparent IGZO based electronic devices (thin-film transistor (TFT), photodetector (PD) and lithium ion battery (LIB)) within a simultaneous processing route on a single glass substrate. While especially IGZO TFTs and PDs are not themselves new, the combination of all transparent elements into a functional and self-powered monolithic unit is an advancement for potential future applications in the realm of Internet of things and consumer electronics. Given the broad scope it is expected that a wider audience would be interested in the current findings, however, there are still a number of open questions that should be resolved:

Response: We really appreciate the reviewer's positive comments on the quality of this manuscript.

1. Within the individual elements of the presented microsystem, the transparent thin-film battery stands out with the highest amount of novelty and innovation. As such, it is suggested to add a little more background information to bring it into context with state-of-the-art literature. In addition, the references 33-35, given as comparison for V₂O₅-based thin-film LIBs are from 2001 to 2004. How about more recent publications?

Response: We have updated the data of the V₂O₅-based thin-film LIBs and a detailed comparison of the V₂O₅-based thin-film LIBs in this work and reported in the literature is summarized in Supplementary Table 6. Based on the comparison, the LIB displays comparable performance to those in the literature in terms of its relatively high specific capacity and good cycling stability.

The revisions have been added on L. 203-205, P. 9 in the manuscript; Supplementary Table 6, P. 24 in the Supplementary Information.

Supplementary Table 6. Performance comparison of the V₂O₅-based thin-film LIB in this work with those in the literature.

Battery configuration	Specific capacity ($\mu\text{Ah cm}^{-2} \mu\text{m}^{-1}$)	Potential window (V)	Current density ($\mu\text{A cm}^{-2}$)	Cycle number (capacity retention)	Ref.
V ₂ O ₅ LiPON Li	33.3	1.5-3.8	30	80 (75%)	11
V ₂ O ₅ LiPON Li	32	2.15-3.8	64	1000 (93.8%)	12
Li _{1.5} V ₂ O ₅ LiPON Li	52	2.15-3.8	10	10 (96.2%)	13
V ₂ O ₅ - Li ₃ PO ₄ LiPON Li	16	0.5-4	2	30 (68.8%)	14
V ₂ O ₅ LiPON Li	54	2.15-3.8	10	60	15
V ₂ O ₅ - Li ₃ PO ₄ LiPON Li	38	0.5-3	0.6	30 (49%)	16
V ₂ O ₅ LiPON Li	20	2.15-3.8	10	50	17
LiV ₂ O ₅ Li ₂ PO ₂ N SnN _x	40	0.5-3.3	50	100	18
V ₂ O ₅ LiPON IGZO	42.6	0.5-3.0	4.3	300 (96%)	This work

References

- Oukassi, S. et al. Ultra-thin rechargeable lithium ion batteries on flexible polymer: design, low temperature fabrication and characterization. *J. Electrochem. Soc.* **164**, A1785-A1791 (2017).
- Xiao, C. F. et al. Ensemble design of electrode-electrolyte interfaces: toward high-performance thin-film all-solid-state Li-metal batteries. *ACS Nano* **15**, 4561-4575

(2021).

13. Navone, C. et al. Lithiated c-V₂O₅ thin-film as positive electrode for rocking-chair solid-state lithium microbattery. *Ionics* **16**, 577-580 (2010).

14. Tsuji, K., Yoshida, M. & Kanno, I. Fabrication of all-solid-state amorphous thin-film Lithium-ion batteries. *PowerMEMS 2021 Virtual Conference* 216-219 (2021).

15. Navone, C., Baddour-Hadjean, R., Pereira-Ramos, J. P. & Salot, R. Sputtered crystalline V₂O₅ thin films for all-solid-state lithium microbatteries. *J. Electrochem. Soc.* **156**, A763-A767 (2009).

16. Kanazawa, S. et al. Deposition and performance of all solid-state thin-film lithium-ion batteries composed of amorphous Si/LiPON/VO-LiPO multilayers. *Thin Solid Films* **697**, 137840 (2020).

17. Oukassi, S., Salot, R. & Pereira-Ramos, J. P. Elaboration and characterization of crystalline RF-deposited V₂O₅ positive electrode for thin film batteries. *Appl. Surf. Sci.* **256**, 149-155 (2009).

18. Pearse, A. et al. Three-dimensional solid-state lithium-ion batteries fabricated by conformal vapor-phase chemistry. *ACS Nano* **12**, 4286-4294 (2018).

2. *The illustration of the microsystem in Fig. 1a shows multiple discrete TFTs and PDs on the substrate whereas for the complete system more interconnects are required between the respective parts. How were such interconnects realized, through external wiring or with a separate deposition of e.g. ITO electrodes?*

Response: In our previous manuscript, the devices were prepared on single substrate but the connections between each device were mainly realized by using a probe station. In order to achieve a fully integrated system, the connections between each device have been realized by depositing ITO interconnects in the revised manuscript. The schematic diagram together with the photograph of the redesigned integrated microsystem is

shown in Fig. 1. The performance of this integrated microsystem is also characterized and displayed in Fig. 5.

The revisions have been added on L. 64-69, P. 3; Fig. 1, P. 4; Fig. 5, P. 16 in the manuscript; Supplementary Fig.1, P. 2 in the Supplementary Information.

Fig. 1. Illustration of the integrated transparent microsystem as well as each component. (a) Schematic diagrams of the integrated microsystem and its components. **(b)** Equivalent circuit diagram of the integrated microsystem. **(c)** Photograph of the complete integrated microsystem. **(d)** Transmittance of each component and the integrated microsystem. The inset is the optical image of the complete integrated microsystem placed onto a logo.

Fig. 5. Collaborative operation of each component in the integrated transparent system. (a) Test setup of LIB charging process by using TFTR as the on-chip rectifier. In this case, the switch S1 is turned-on and the switch S2 is turned-off. LIB charging process under AC sinusoidal input signals with **(b)** different voltage amplitudes and **(c)** different frequencies. **(d)** Test setup of PD photoresponse by using the charged LIB as the on-chip power. In this case, the switch S1 is turned-off and the switch S2 is turned-on. **(e)** PD photoresponse under light illumination with different power densities at a fixed wavelength of 405 nm. **(f)** Evaluation of operating time for the charged LIB used in the test setup displayed in Fig. 5d.

3. While the process parameters for each respective layers are well documented in the Supporting Information, neither there or in the main manuscript can information be found on the actual device dimensions that were used in the study (e.g. TFT channel length/width, etc). This information needs to be included for all components as well as the substrate.

Response: The geometrical sizes of each device, including the channel width ($\sim 300 \mu\text{m}$) and length ($\sim 30 \mu\text{m}$) of the TFT, the resistance width ($\sim 500 \mu\text{m}$) and length ($\sim 150 \mu\text{m}$) of the PD, the effective area (defined as the overlap between the anode and cathode current collectors, width \times length = $3.8 \text{ mm} \times 3.0 \text{ mm}$) of the thin-film LIB,

and the size (width \times length = 8 mm \times 8 mm) of the substrate, have been added in the revised manuscript.

The revisions have been added on L. 420-429, P. 18; L. 452, P.19 in the manuscript.

4. Some questions regarding the TFTs:

i. Did the devices show any hysteresis? Please include forward and backward sweep in 5a).

Response: As seen in Fig. 4a below, the TFT displays a hysteresis of 0.4 V. Note that the hysteresis is defined by the threshold-voltage difference of the transfer curves under forward and backward sweepings.

Fig. 4. Characterization of the TFT, TFTR and PD at $P_{O_2} = 0.04$ Pa. (a) Transfer curve, (b) output characteristics, and (c) rectification curve of the TFT device. (d) AC sinusoidal signals generated by a signal generator as the TFTR input. (e) Half-wave DC output signals of the TFTR under different-amplitude AC sinusoidal in put signals with a fixed frequency of 100 Hz. (f) Half-wave

DC output signals of the rectifier under different-frequency AC sinusoidal input signals with a fixed amplitude of 7 V. (g) Dependence of photocurrent and responsivity on the 405-nm light intensity for the PD device. (h) Current-time curve measured under a 405-nm light intensity of 2.3 mW cm^{-2} at a bias of 5 V for the PD device. (i) High-resolution current-time curve under a 405-nm light intensity of 2.3 mW cm^{-2} at a bias of 5 V used for measuring the rise time and decay time of the PD device.

The revisions have been added on L. 250-254, P. 11; Fig. 4a, P. 12 in the manuscript; Supplementary Table 4, P. 22 in the Supplementary Information.

ii. What was the capacitance of the dielectric and at what frequency was it measured at?

Response: The capacitance of the HfLaO dielectric is measured to be $0.17 \mu\text{F cm}^{-2}$ by using an impedance analyzer (Agilent 4294A) at 1 MHz, as seen in Supplementary Fig. 12. The k value of the dielectric is calculated to 14.5. Also. The capacitor displays a low leakage ($\sim 10^{-5} \text{ A cm}^{-2}$ at -1 V).

Supplementary Fig. 12. Capacitance and leakage as a function of applied voltage for the HfLaO-based capacitor. The capacitance is measured by using an impedance analyzer at 1 MHz and the leakage is measured by using a semiconductor parameter analyzer.

The revisions have been added on L. 246-249, P. 11; L. 487-490, P. 20 in the manuscript; Supplementary Fig. 12, P. 13 in the Supplementary Information.

iii. The off-current at negative V_g is rather high. What is believed to be the limiting factor here? Is it related to the IGZO channel material or does the dielectric play a role? Including the gate current in Fig. 5a) could help to clarify this.

Response: The low on/off current ratio ($I_{on}/I_{off} \sim 10^3$) should be mainly ascribed to metallic contamination induced during depositing the TFT films since the transistor is quite sensitive to the metallic contamination. In order to validate this analysis, we have re-prepared the TFT device based on the same fabrication processes. The main difference between the re-prepared TFT and previous TFT is that the dielectric and IGZO films are deposited by using a ‘clean’ sputter, which is only used for dielectric and semiconductor deposition (not allowed for metal deposition) and thus the metallic contamination can be effectively suppressed. As shown in Fig. R3 below, the re-prepared TFT displays better performance than the previous one in terms of its higher I_{on}/I_{off} ratio (1.4×10^6 vs. 3.4×10^3), higher carrier mobility ($23.3 \text{ cm}^2 \text{ V}^{-1} \text{ s}^{-1}$ vs. $4.5 \text{ cm}^2 \text{ V}^{-1} \text{ s}^{-1}$) and smaller sub-threshold swing ($209.2 \text{ mV dec}^{-1}$ vs. $434.3 \text{ mV dec}^{-1}$). It is noted that the main contribution of this manuscript is to develop an IGZO-based transparent LIB and integrated transparent microsystem. The above results also suggest that it has potential to further improve the performance of this integrated microsystem by optimizing the fabrication conditions.

The revisions have been added on L. 430-435, P. 18; Fig. 4a-c, P. 12 in the manuscript; Supplementary Fig. 1, P. 2; Supplementary Table 1, P. 19 in the Supplementary Information.

Fig. R3. Transfer curves of the TFTs **(a)** in the current work and **(b)** in the previous work.

iv. Although a good Ohmic contact between IGZO and ITO was remarked on by the authors, there still seems to be a noticeable S-shape for V_{ds} values close to 0 V, that is typically indicative of contact resistance. A comment on this behavior would be appreciated.

Response: The output characteristics of the re-prepared TFT are shown in Fig. 4b above and the crowding phenomenon at low V_{DS} , which is usually associated with large contact resistance, is not observed. In addition, a relatively high driving output current can be obtained for this device ($I_{DS} \sim 10.5 \mu\text{A}$ at $V_{DS} = 5 \text{ V}$ and $V_{GS} = 5 \text{ V}$).

The revisions have been added on L. 258-260, P. 11; Fig. 4b, P. 12 in the manuscript.

5. It is somewhat surprising to see an IGZO photoresponse at a wavelength of 660 nm. Could the authors please elaborate on this part? In other studies (compare e.g. Kim et al., *J Mater Chem*, 8, 165-172, 2020, <https://doi.org/10.1039/C9TC04982G>) defects that create subgap states are specifically introduced into IGZO to make this possible.

Response: The IGZO photodetector exhibits a weak photoresponse at a wavelength of 660 nm (the photon energy is 1.9 eV for 660-nm light). This should be ascribed to the oxygen vacancy in the IGZO film, which would create subgap states in the IGZO

bandgap (D. Kim, *et al.*, *J. Mater. Chem. C*, 8, 165, 2020.). These subgap states are located at about 1.7 eV below the conduction band minimum given that the IGZO bandgap is 3.2 eV (H. Qian, *et al.*, *J. Phys. D: Appl. Phys.*, 49, 395104, 2016; T. Kamiya, *et al.*, *Sci. Technol. Adv. Mater.*, 11, 044305, 2010.) and thus can make a photoresponse under light illumination with a wavelength of 660 nm.

The revisions have been added on L. 331-333, P. 14; L. 334-336, P. 15 in the manuscript.

6. *Metal oxide semiconductors can suffer from a persistent photocurrent after illumination (compare e.g. Jeon et al., Nat. Mater., 11, 301-305, 2012, <https://doi.org/10.1038/nmat3256>). Was any special precaution taken or procedure carried out to prevent this?*

Response: Light illumination would induce charge trapping in the defects for metal-oxide semiconductors, thus leading to the persistent photocurrent. Similar phenomenon also occurs in our IGZO photodetector, as seen in Fig. 4h and i above. Based on the literature, two main methods can be used to suppress this persistent photocurrent: (i) passivating the defects in the IGZO film (S. Park, *et al.*, *ACS Appl. Electron. Mater.*, 1, 2655, 2019.); (ii) applying a gate bias on the device to compensate this persistent photocurrent (S. Jeon *et al.*, *Nat. Mater.*, 11, 301, 2012; H. Liu *et al.*, *IEEE Electron Devices Lett.*, 43, 1247, 2022.).

The revisions have been added on L. 345-350, P. 15 in the manuscript.

7. *The transparency of the full microsystem is given as larger than 76 % at a wavelength of over 700 nm. This is borderline in the visible spectrum and for a representative quantification it is recommended to either pick a more central wavelength (e.g. 550 nm) or to compute an average over the visible spectrum.*

Response: According to the reviewer's suggestion, the transmittance (> 66% at 550-

nm light) at the central wavelength of the visible light has been used in the revised manuscript.

The revisions have been added on L. 78, P. 4 in the manuscript.

8. The whole fabrication scheme was kept to room temperature depositions with the aim to keep the active films amorphous and thus to reduce nonuniformities arising from grain boundaries. Adding additional device data related to uniformity and/or stability would strengthen this approach.

Response: For polycrystalline materials, many defects exist in their grain boundaries, which would cause instability issues of the corresponding devices. Moreover, these grain boundaries are randomly distributed, thus resulting in nonuniformities of the corresponding devices (*J. Troughton, et al., J. Mater. Chem. C, 7, 12388, 2019.*). Amorphous IGZO is effective to solve the issues caused by the grain boundaries and also has high carrier mobility and high optical transparency, thus having received much attention since it was invented in 2003 (*K. Nomura, et al., Science, 300, 1269, 2003.*). This is also the reason that we choose IGZO to construct the integrated system. Other active films used in this work are also prepared with an amorphous state for retaining good uniformities of the microsystem. Performance comparison between the devices constructed by amorphous and polycrystalline IGZO respectively is persuasive to demonstrate the advantages of using amorphous IGZO. However, the IGZO crystalline temperature is about 600 °C (*A. Suko, et al., Jpn. J. Appl. Phys., 55, 035504, 2016.*). Both the ITO electrode and glass substrate cannot endure this temperature. Therefore, the above performance comparison is not performed in this work.

The revisions have been added on L. 80-82, P. 4 in the manuscript.

9. What was the measurement environment? Since a cryogenic chamber was mentioned, were the tests carried out under vacuum?

Response: In order to suppress the influences of the atmosphere on the device performance, the measurements were carried out at a vacuum environment with a pressure of 10^{-3} mbar.

The revisions have been added on L. 493-494, P. 20 in the manuscript.

10. Are there any estimations for how long the microsystem could be operated on a single battery charge?

Response: In order to estimate the operating time, the LIB is firstly charged by using the setup diagram shown in Fig. 5a above, in which an AC sinusoidal signal with a frequency of 300 Hz and amplitude of 7 V created by a signal generator is applied to charge the LIB. The charging process is terminated after charging for 200 s. After that, as seen in the inset of Fig. 5f, the LIB is series connected to an external load resistance R_{load} by using a probe station. This load resistance is used to mimic the PD photoresistance and has similar value ($\sim 3 \text{ M}\Omega$) to the PD photoresistance under light illumination with a wavelength of 405 nm and a power density of 3.4 mW cm^{-2} . Fig. 5f shows the corresponding LIB discharging curve as the LIB voltage decreases to a cut-off voltage of 0.5 V and the operating time is 1482 s in this condition.

The revisions have been added on L. 389-397, P. 17; Fig. 5f, P. 16 in the manuscript.

Reviewer #4:

The manuscript entitled “Integration of microbattery with thin-film electronics for constructing an integrated and transparent microsystem based on InGaZnO”, by Bin Jia, et al. presents transparent IGZO based batteries, TFTs, and a light sensor on a glass substrate. The paper is well written and shows the very relevant and important realization of a transparent thin-film power source for wearable systems. At the same time, the manuscript is a bit unclear about the integration of all components (based in the current version I assume they are actually not integrated into a system). Hence, I see the following options:

- Focus on the battery part - which is novel, and the related material/device characterization. Then submit this work to a more specialized journal.*
- If it is true, that the individual components were not connected on the substrate I would not consider this a microsystem. This would make it difficult to publish the work in NatCom*
- If there is a fully integrated system, or if there will be one after some revision, I would support accepting the work, if also the following comments are addressed.*

Response: We really appreciate the reviewer’s positive comments on this manuscript. In our previous version, the devices were prepared on single substrate but the connections between each device were mainly realized by using a probe station. According to the reviewer’s suggestion, the connections between each device have been realized by depositing ITO interconnects for achieving a fully integrated microsystem in the revised manuscript. The schematic diagram together with the photograph of the redesigned integrated microsystem is shown in Fig. 1. Also, the performance of this redesigned integrated microsystem is characterized and displayed in Fig. 5.

The revisions have been added on L. 64-69, P. 3; L. 78-82, P. 4; L. 357-362, P. 15; L. 363-370, P. 16; L. 382-397, P. 17; Fig. 1, P. 4; Fig. 5, P. 16 in the manuscript; Supplementary Fig. 1, P. 2; Supplementary Table 1, P. 19 in the Supplementary Information.

Fig. 1. Illustration of the integrated transparent microsystem as well as each component. (a) Schematic diagrams of the integrated microsystem and its components. **(b)** Equivalent circuit diagram of the integrated microsystem. **(c)** Photograph of the complete integrated microsystem. **(d)** Transmittance of each component and the integrated microsystem. The inset is the optical image of the complete integrated microsystem placed onto a logo.

Fig. 5. Collaborative operation of each component in the integrated transparent system. (a) Test setup of LIB charging process by using TFTR as the on-chip rectifier. In this case, the switch S1 is turned-on and the switch S2 is turned-off. LIB charging process under AC sinusoidal input signals with **(b)** different voltage amplitudes and **(c)** different frequencies. **(d)** Test setup of PD photoresponse by using the charged LIB as the on-chip power. In this case, the switch S1 is turned-off and the switch S2 is turned-on. **(e)** PD photoresponse under light illumination with different power densities at a fixed wavelength of 405 nm. **(f)** Evaluation of operating time for the charged LIB used in the test setup displayed in Fig. 5d.

Additional more technical comments are:

1. Fig 1e quantifies the transmittance, but in addition I would expect to see a much higher resolution and clearer photograph of the system.

Response: Fig. 1 above shows the schematic diagram and corresponding photograph of this integrated system.

The revisions have been added on Fig. 1c, P. 4 in the manuscript.

2. The novelty of Fig. 2 compared to the state of the art in the context of IGZO is very

limited. However, it could be part of the supplementary information.

Response: Fig. 2 (corresponding to Supplementary Fig. 2 in the revised manuscript) has been reorganized and moved to the Supplementary Information.

The revisions have been added on Supplementary Fig. 2, P.3 in the Supplementary Information.

3. *The craterisation of the battery is not fully consistent. E.g. it would be better to show a full set of numbers and graphs for the same current densities.*

Response: Specific capacity, cycling stability and rate performance are three main parameters for the lithium-ion batteries (LIBs). Specific capacity is usually characterized at a relatively low specific current (0.05 A g⁻¹ in this work) for making the electrochemical reactions sufficient. The characterization of cycling stability is time-consuming (e.g., it takes the IGZO LIBs about 16 days to complete 250 cycling), and thus a relatively high specific current (0.4 A g⁻¹ in this work) is adopted to accelerate the cycling test. Rate performance is used to measure how fast the LIBs can charge and discharge, therefore, a specific current as high as 4.0 A g⁻¹ is used in this work. In order to make a fair comparison between the samples, each electrochemical characterization is performed at the same specific current for each sample.

The revisions have been added on L. 136, P.7; L. 165, P. 8 in the manuscript..

4. *Transistors:*

i. *The achieved performance needs more discussion, in particular the current ratio is not necessarily in line with the state of the art.*

Response: The low on/off current ratio ($I_{on}/I_{off} \sim 10^3$) should be mainly ascribed to metallic contamination during depositing the TFT films since the transistor is quite sensitive to the metallic contamination. In order to validate this analysis, we have re-prepared the TFT device based on the same fabrication processes. The main difference

between the re-prepared TFT and previous TFT is that the dielectric and IGZO films are deposited by using a ‘clean’ sputter, which is only used for dielectric and semiconductor deposition (not allowed for metal deposition) and thus the metallic contamination can be effectively suppressed. As shown in Fig. R3 below, the re-prepared TFT displays better performance than the previous one in terms of its higher I_{on}/I_{off} ratio (1.4×10^6 vs. 3.4×10^3), higher carrier mobility ($23.3 \text{ cm}^2 \text{ V}^{-1} \text{ s}^{-1}$ vs. $4.5 \text{ cm}^2 \text{ V}^{-1} \text{ s}^{-1}$) and smaller sub-threshold swing ($209.2 \text{ mV dec}^{-1}$ vs. $434.3 \text{ mV dec}^{-1}$). It is noted that the main contribution of this manuscript is to develop an IGZO-based transparent LIB and integrated transparent microsystem. The above results also suggest that it has potential to further improve the performance of this integrated microsystem by optimizing the fabrication conditions.

The revisions have been added on L. 430-435, P. 18; Fig. 4a-c, P. 12 in the manuscript; Supplementary Fig. 1, P. 2; Supplementary Table 1, P. 19 in the Supplementary Information.

Fig. R3. Transfer curves of the TFTs **(a)** in the current work and **(b)** in the previous work.

ii. The text mentions good ohmic contacts, but Fig 5b actually looks like there is significant impact of the contact resistance at low V_{ds} (although it is hard to say as the resolution is a bit low).

Response: The output characteristics of the re-prepared TFT are shown in Fig. 4b below and the crowding phenomenon at low V_{DS} , which is usually associated with large

contact resistance, is not observed. In addition, this TFT presents a relatively high driving output ($I_{DS} \sim 10.5 \mu\text{A}$ at $V_{DS} = 5 \text{ V}$ and $V_{GS} = 5 \text{ V}$).

The revisions have been added on L. 258-260, P. 11; Fig. 4b, P. 12 in the manuscript.

Fig. 4. Characterization of the TFT, TFTR and PD at $P_{O_2} = 0.04 \text{ Pa}$. (a) Transfer curve, (b) output characteristics, and (c) rectification curve of the TFT device. (d) AC sinusoidal signals generated by a signal generator as the TFTR input. (e) Half-wave DC output signals of the TFTR under different-amplitude AC sinusoidal input signals with a fixed frequency of 100 Hz. (f) Half-wave DC output signals of the rectifier under different-frequency AC sinusoidal input signals with a fixed amplitude of 7 V. (g) Dependence of photocurrent and responsivity on the 405-nm light intensity for the PD device. (h) Current-time curve measured under a 405-nm light intensity of 2.3 mW cm^{-2} at a bias of 5 V for the PD device. (i) High-resolution current-time curve under a 405-nm light intensity of 2.3 mW cm^{-2} at a bias of 5 V used for measuring the rise time and decay time of the PD device.

iii. Fig.5a needs more details such as the W/L ratio, the gate leakage current, the current in the linear region, etc..

Response: Besides the W/L (300 μm /30 μm) ratio of the TFT device, critical sizes of the other devices, including the resistance width ($\sim 500 \mu\text{m}$) and length ($\sim 150 \mu\text{m}$) of the PD and the effective area (defined as the overlap between the anode and cathode current collector, width \times length = 3.8 mm \times 3.0 mm) of the LIB, etc, have also been added in the revised manuscript. In addition, as seen in Supplementary Fig 12, the areal leakage and capacitance densities of the gate dielectric have been added. The dielectric displays a relatively low leakage ($\sim 10^{-5} \text{ A cm}^{-2}$ at -1V). The corresponding k value of the dielectric is calculated to be 14.5.

The revisions have been added on L. 246-249, P. 11; L. 420-429, P. 18; L. 487-490, P. 20 in the manuscript; Supplementary Fig. 12, P.13 in the Supplementary Information.

Supplementary Fig. 12. Capacitance and leakage as a function of applied voltage for the HfLaO-based capacitor. The capacitance is measured by using an impedance analyzer at 1 MHz and the leakage is measured by using a semiconductor parameter analyzer.

iv. The manuscript mentions the turn-on voltage and then treats it like the threshold voltage, however these are not the same. Please clarify this.

Response: The turn-on voltage ($V_{\text{turn-on}}$) is equal to the threshold voltage (V_{th}) and has

been corrected to V_{th} in the revised manuscript. As seen in Fig. 4a, the V_{th} is obtained from the intercept of $I_D^{1/2}$ vs. V_G in the saturation region with the x-axis.

The revisions have been added on L. 278, P. 13; Fig. 4a, P. 12 in the manuscript.

v. Figs 5d and 5e should use the same scale, and fig 5f should also show a scale (even if there is a vertical offset used, the difference in voltage for all measurements should be shown).

Response: Fig. 5d, e and f (corresponding to Fig. 4d, e and f in the revised manuscript) have been redrawn according to the reviewer's suggestion.

The revisions have been added on Fig. 4d-f, P. 12 in the manuscript.

5. *Photo detectors:*

• *Effectively this device is a resistive sensor not a photodiode so it does not make a lot of sense to show the current (as it depend on the applied voltage – which is not given for al measurements). Here the resistance variation of the device should be shown.*

Response: In Fig. 4h and i as shown above, a constant voltage bias (~ 5 V) applied by a Keithley 4200-SCS semiconductor parameter analyzer is used for the photodetector measurements. For comparison, in Fig. 5e, the voltage applied by the LIB is used for the photodetector measurements. Because the LIB voltage decreases with working time, the photoresponse characteristics of the PD device is described by using the resistance variation.

The revisions have been added on L. 271-272, P. 12; L. 302-305, P. 13; L. 322-324, P.14; L. 385-389, P. 17; Fig. 5e, P. 16 in the manuscript.

6. *Please give the dimensions of all devices.*

Response: The dimensions of each device, including the channel width (~ 300 μm) and

length ($\sim 30 \mu\text{m}$) of the TFT, the resistance width ($\sim 500 \mu\text{m}$) and length ($\sim 150 \mu\text{m}$) of the PD and the effective area (defined as the overlap between the anode and cathode current collectors, width \times length = $3.8 \text{ mm} \times 3.0 \text{ mm}$) of the LIB, etc, have been added in the revised manuscript.

The revisions have been added on L. 420-429, P. 18 in the manuscript.

7. Integration:

i. Specify how the devices were connected in Fig 5.

Response: As discussed earlier, the microsystem has been re-designed and prepared for achieving a fully integrated microsystem and the details are shown in Fig 1 above. In this microsystem, the connections of each device are realized by using ITO interconnects.

The revisions have been added on L. 67-68, P. 3; Fig. 1a and c, P. 4 in the manuscript.

ii. How were the switches S1 and S2 implemented?

Response: The working states of the switches S1 and S2 are controlled by changing the gate voltage (V_G): when a V_G of 6 V is applied on the gate by using a Keithley 4200-SCS semiconductor parameter analyzer, the switches are turned-on; for comparison, when the gate is floating, the switches are turned-off.

The revisions have been added on L. 357-359, P. 15; L. 382-383, P. 17; Fig. 5a and c, P. 16 in the manuscript.

iii. Please show a micrograph of the full system.

Response: As seen in Fig. 1a and c above, the schematic diagram as well as the

photograph of the redesigned full system is presented.

The revisions have been added on Fig. 1c, P. 4 in the manuscript.

iv. It is written that after charging the battery the AC input signal was “shut down”. Please specify this, was the input 0V or floating? Was it still connected to the TFT? This would most probably influence the discharge.

Response: As seen in Fig. 5, the AC input signal is disconnected to the TFT after charging the battery. A clear description has been added in the revised manuscript.

The revisions have been added on L. 364-365, P. 16 in the manuscript.

REVIEWERS' COMMENTS

Reviewer #1 (Remarks to the Author):

InGaZnO (IGZO) is an important semiconductor for widely studied TFTs and photodetectors. Although there is still lack of transparent thin-film lithium-ion battery (LIB) with IGZO as the anode, the single components InOx, GaOx, ZnOx have been widely used in LIB. In fact, every oxide could be/have been used as the anode of LIBs. Theoretically, IGZO could be used as the anode of LIBs. If it could not, the authors should provide some proofs.

1. Could the authors provide the more details or results on the reasons of IGZO as the anode of LIBs? Could IGZO provide better LIB performance than Binary or ternary counterpart, such as InOx, GaOx and/or ZnOx? Or just try IGZO as the anode of LIBs.

2. Which is better used as the anode of LIBs, IGZO or ITO? For the anode of LIBs, what is the function of every element of IGZO?

3. "IGZO films with different oxygen contents are prepared by changing the oxygen partial pressure ($PO_2 = 0$ Pa, 0.04 Pa and 0.15 Pa) during sputtering."
The oxygen partial pressure is a very important parameter for IGZO film. The authors selected 0 Pa, 0.04 Pa and 0.15 Pa. However, the range between 0.04 to 0.15 Pa may be too large. The results showed best at 0.04 Pa. Perhaps the best results may lie at 0.08, 0.12 or other Pa.

4. "each device displays an acceptable performance and collaboratively works well."
The high performance is also very important for the significance of paper. The authors should provide some important device performance parameters in the abstract, Result Discussion and/or Discussion.
It is better to have performance comparable to or better than most previous work.

5. The TFT performance at $PO_2 = 0.15$ Pa may have some problem. Its mobility is the lowest, and On-off current ratio is also the lowest. What is the reason?

Reviewer #2 (Remarks to the Author):

The revised manuscript has provided deeper purpose analyses of this proposed transparent system which make this work more understandable and agreeable. The theoretical capacity ($mA\ h\ g^{-1}$) of IGZO metal oxide as electrodes for metal ion batteries has been estimated carefully and the extensive discussion on the difference between experimental results and theoretical analyses has been made. Thus, the revised manuscript shows that the IGZO anode film in this work presented comparable performance to the reported metal-oxide anode films in the literature. The results of collaborative work of lithium-ion battery (LIB), thin-film transistor (TFT) and photodetector (PD) as the transparent microsystem had been well described to guaranty the feasibility of this proposed system.

Reviewer #3 (Remarks to the Author):

The authors have made a substantial effort to add more background information, material characterization and device data in order to address the reviewers comments. Most importantly, they showed the actual realization of the combined system with interconnects on the same chip. In total, the quality of the manuscript has improved and its publication can now be recommended.

Reviewer #4 (Remarks to the Author):

The revised version of the manuscript entitled "Integration of microbattery with thin-film electronics for constructing an integrated and transparent microsystem based on InGaZnO", by Bin Jia, et al. addresses my comments in a proper way. Most importantly the new version of the system is indeed fully integrated. I now support accepting the paper.

I only have the following very minor comments which could help improving the final version of the paper.

1. Adding a scale bar to Fig 1c would be beneficial.
2. It is good that the gate leakage current was added to the manuscript, however it would be much easier more initiative to rate the quality of the gate insulator if the gate current would be shown in Fig 4a (using the same y axis scaling).
3. One remark about the notation: The drain current is called I_{DS} – this does not make sense as, opposed to voltages, currents are only defined in a single point so what is shown is either the drain current I_D or the source current I_S but not both. The figures should be updated accordingly.

Point-by-point response to the reviewers' comments

Reviewer #1

InGaZnO (IGZO) is an important semiconductor for widely studied TFTs and photodetectors.

Although there is still lack of transparent thin-film lithium-ion battery (LIB) with IGZO as the anode, the single components InO_x , GaO_x , ZnO_x have been widely used in LIB.

In fact, every oxide could be/have been used as the anode of LIBs.

Theoretically, IGZO could be used as the anode of LIBs. If it could not, the authors should provide some proofs.

Response: A full integration of various kinds of components (including electronic devices, sensing devices and energy devices, etc.) onto a single substrate to construct an integrated microsystem can significantly improve the functions of transparent electronics. Moreover, constructing an integrated microsystem based on the same functional material is helpful and desirable to enhance the microsystem compactness and simplify its fabrication process. As the reviewer mentioned, IGZO is an important functional material and has been widely used in thin-film transistors (TFTs) and photodetectors (PDs). Therefore, we are motivated to develop an IGZO-based LIB for realizing a fully integrated transparent microsystem. Both the theoretical calculation (the details are seen in Supplementary Note 1) and experimental results (Fig. 2 and Fig. 3 in the manuscript) suggest that IGZO is suitable as the LIB anode.

The revisions have been added on L. 47-49, P. 3 in the manuscript.

1. Could the authors provide the more details or results on the reasons of IGZO as the anode of LIBs? Could IGZO provide better LIB performance than Binary or ternary counterpart, such as InO_x , GaO_x and/or ZnO_x ? Or just try IGZO as the anode of LIBs.

Response: In order to realize a compact microsystem and simplify its fabrication process, it is desirable to construct an integrated microsystem based on the same

functional material. Due to good uniformity, high carrier mobility and high optically transparency, IGZO has received much attention and has been widely used in transparent electronic devices (e.g. TFT) and transparent sensors (e.g. PD), respectively. Also, IGZO has already been adopted in some electronic products (e.g. iPad Air) in recent years, further demonstrating its great potential in electronics. However, there is still lack of an IGZO-based transparent LIB for constructing a fully integrated transparent microsystem. Therefore, this work investigates IGZO as the LIB anode although many other metal-oxide anode films (e.g. In_2O_3 , ZnO and NiO) have been reported in the literature. By optimizing its oxygen content, the IGZO anode film displays comparable performance compared with other metal-oxide anode films reported in the literature (the details are seen in Supplementary Table 5).

The revisions have been added on L. 47-49, P. 3 in the manuscript.

2. Which is better used as the anode of LIBs, IGZO or ITO? For the anode of LIBs, what is the function of every element of IGZO?

Response: As discussed above, the motivation of investigating IGZO as the LIB anode is to construct an integrated microsystem (including LIB, TFT and PD) based on the same functional material. The effects of oxygen content (or oxygen vacancies) on the IGZO anode performance have been investigated in this work by varying oxygen partial pressure (P_{O_2}) during IGZO preparation. It is found that both the ionic and electrical conductivities of the IGZO anode are dependent on the oxygen content. The IGZO anode at $P_{\text{O}_2}=0.04$ Pa displays better performance than the ones at $P_{\text{O}_2}=0$ Pa and $P_{\text{O}_2}=0.15$ Pa mainly because of its higher ionic conductivity (vs. the anode at $P_{\text{O}_2} = 0$ Pa) and higher electrical conductivity (vs. the anode at $P_{\text{O}_2} = 0.15$ Pa) as demonstrated in the manuscript. The effects of cations on the IGZO properties have been reported in the literature (*H. Hosono, J. Non Cryst. Solids, 352, 851, 2006; E. Fortunato, et al., Adv. Mater. 24, 2945, 2012; X. Tang et al., ACS Appl. Mater. Interfaces, 10, 5519, 2018.*): indium cation is the main contributor to the carrier mobility (and thus the electrical conductivity) of IGZO; gallium cation is effective to suppress electron carrier

generation due to its high ionic potential. Besides, it has been reported that gallium cation is helpful to improve the battery stability because of its low melting point; zinc cation contributes to the amorphous structure of IGZO, which is desirable to improve the IGZO uniformity.

3. *“IGZO films with different oxygen contents are prepared by changing the oxygen partial pressure ($P_{O_2} = 0 \text{ Pa}$, 0.04 Pa and 0.15 Pa) during sputtering.”*

The oxygen partial pressure is a very important parameter for IGZO film. The authors selected 0 Pa , 0.04 Pa and 0.15 Pa . However, the range between 0.04 to 0.15 Pa may be too large. The results showed best at 0.04 Pa . Perhaps the best results may lie at 0.08 , 0.12 or other Pa .

Response: We have investigated the effects of IGZO prepared at various P_{O_2} on the device performance. As the reviewer suggested, the range between 0.04 Pa and 0.15 Pa is large and a more careful optimization is worthy to be performed in the near future.

The revisions have been added on L. 409-410, P. 15 in the manuscript.

4. *“each device displays an acceptable performance and collaboratively works well.”*

The high performance is also very important for the significance of paper.

The authors should provide some important device performance parameters in the abstract, Result Discussion and/or Discussion.

It is better to have performance comparable to or better than most previous work.

Response: According to the reviewer’s suggestion, one typically key parameter of each device is added in the abstract (note that not all the key parameters are provided in the abstract due to the limitation of word counts). The key parameters of each device are summarized in Supplementary Table 4. The performance comparison is displayed in Supplementary Table 5 and Supplementary Table 6, both of which demonstrate the merit of using IGZO as the LIB anode film.

The revisions have been added on L. 20, 21, 22, P. 2 in the manuscript.

5. The TFT performance at $P_{O_2} = 0.15$ Pa may have some problem. Its mobility is the lowest, and On-off current ratio is also the lowest. What is the reason?

Response: The TFT on-current and off-current tend to increase and decrease respectively with increasing P_{O_2} . The TFT at $P_{O_2} = 0.15$ Pa presents much lower carrier mobility ($8.3 \text{ cm}^2 \text{ V}^{-1} \text{ s}^{-1}$ vs. $58.0 \text{ cm}^2 \text{ V}^{-1} \text{ s}^{-1}$ at $P_{O_2} = 0$ Pa and $23.3 \text{ cm}^2 \text{ V}^{-1} \text{ s}^{-1}$ at $P_{O_2} = 0.04$ Pa) and thus smaller on-current than the devices at $P_{O_2} = 0$ Pa and $P_{O_2} = 0.04$ Pa, which contributes to its lowest on-off current ratio.

The revisions have been added on L. 244-245, P. 9; L. 246, P. 10 in the manuscript.

Reviewer #2

The revised manuscript has provided deeper purpose analyses of this proposed transparent system which make this work more understandable and agreeable. The theoretical capacity (mA h g^{-1}) of IGZO metal oxide as electrodes for metal ion batteries has been estimated carefully and the extensive discussion on the difference between experimental results and theoretical analyses has been made. Thus, the revised manuscript shows that the IGZO anode film in this work presented comparable performance to the reported metal-oxide anode films in the literature. The results of collaborative work of lithium-ion battery (LIB), thin-film transistor (TFT) and photodetector (PD) as the transparent microsystem had been well described to guaranty the feasibility of this proposed system.

Response: We really appreciate the reviewer's positive comments on the quality of this manuscript.

Reviewer #3

The authors have made a substantial effort to add more background information, material characterization and device data in order to address the reviewers comments. Most importantly, they showed the actual realization of the combined system with interconnects on the same chip. In total, the quality of the manuscript has improved and its publication can now be recommended.

Response: We really appreciate the reviewer's positive comments on the quality of this manuscript.

Reviewer #4:

The revised version of the manuscript entitled *Integration of microbattery with thin-film electronics for constructing an integrated and transparent microsystem based on InGaZnO*”, by Bin Jia, et al. addresses my comments in a proper way. Most importantly the new version of the system is indeed fully integrated. I now support accepting the paper.

Response: We really appreciate the reviewer’s positive comments on the quality of this manuscript.

I only have the following very minor comments which could help improving the final version of the paper.

1. Adding a scale bar to Fig 1c would be beneficial.

Response: The scale bar has been added in Fig. 1c.

The revisions have been added on Fig. 1c, P. 24 in the manuscript.

Fig. 1. Illustration of the integrated transparent microsystem as well as each component. (a)

Schematic diagrams of the integrated microsystem and its components. **(b)** Equivalent circuit diagram of the integrated microsystem. **(c)** Photograph of the complete integrated microsystem. **(d)** Transmittance of each component and the integrated microsystem. The inset is the optical image of the complete integrated microsystem placed onto a logo.

2. It is good that the gate leakage current was added to the manuscript, however it would be much easier more initiative to rate the quality of the gate insulator if the gate current would be shown in Fig 4a (using the same y axis scaling).

Response: The gate leakage has been shown in Fig. 4a by using the same y-axis scaling.

The revisions have been added on Fig. 4a, P. 27 in the manuscript.

Fig. 4. Characterization of the TFT, TFTR and PD at $P_{O_2} = 0.04$ Pa. **(a)** Transfer curve, **(b)** output characteristics, and **(c)** rectification curve of the TFT device. The inset of Fig. 4a shows areal capacitance (C_i , $\mu\text{F cm}^{-2}$) and leakage (I_i , A) as a function of applied voltage for the HfLaO-based capacitor. **(d)** AC sinusoidal signals generated by a signal generator as the TFTR input. **(e)** Half-

wave DC output signals of the TFTR under different-amplitude AC sinusoidal input signals with a fixed frequency of 100 Hz. **(f)** Half-wave DC output signals of the rectifier under different-frequency AC sinusoidal input signals with a fixed amplitude of 7 V. **(g)** Dependence of photocurrent and responsivity on the 405-nm light intensity for the PD device. **(h)** Current-time curve measured under a 405-nm light intensity of 2.3 mW cm^{-2} at a bias of 5 V for the PD device. **(i)** High-resolution current-time curve under a 405-nm light intensity of 2.3 mW cm^{-2} at a bias of 5 V used for measuring the rise time and decay time of the PD device.

3. *One remark about the notation: The drain current is called I_{DS} – this does not make sense as, opposed to voltages, currents are only defined in a single point so what is shown is either the drain current I_D or the source current I_S but not both. The figures should be updated accordingly.*

Response: I_{DS} has been corrected to I_D in the revised manuscript.

The revisions have been added on L. 247, P. 10; Fig. 4a and b, P. 27 in the manuscript; Supplementary Fig. 12. P. 13 in the Supplementary Information.

Supplementary Fig. 12. Transfer curves of the TFTs with IGZO channel layer prepared at **(a)** $P_{O_2} = 0 \text{ Pa}$ and **(b)** $P_{O_2} = 0.15 \text{ Pa}$. The V_{th} is obtained from the intercept of $I_D^{1/2}$ vs. V_G in the saturation region with the x -axis. ΔV_{th} is defined by the threshold-voltage difference of the transfer curves under forward and backward sweepings.